# Remote Sensing and Modeling of the Cryosphere in High Mountain Asia: A Multidisciplinary Review

Qinghua Ye [1,2], Yuzhe Wang [3], Lin Liu [4], Linan Guo [1], Xueqin Zhang [2], Liyun Dai [5], Limin Zhai [6,7], Yafan Hu [1,7], Nauman Ali [1,7], Xinhui Ji [1,7], Youhua Ran [5], Yubao Qiu [8], Lijuan Shi [8], Tao Che [5], Ninglian Wang [9], Xin Li [1] and Liping Zhu [1,*]

1   State Key Laboratory Tibetan Plateau Earth System, Environment and Resources (TPESER), Institute of Tibetan Plateau Research, Chinese Academy of Sciences, Beijing 100101, China; yeqh@itpcas.ac.cn (Q.Y.); guoln@itpcas.ac.cn (L.G.); huyafan@itpcas.ac.cn (Y.H.); naumanali@itpcas.ac.cn (N.A.); jixinhui@itpcas.ac.cn (X.J.); xinli@itpcas.ac.cn (X.L.)
2   Institute of Geographic Sciences and Natural Resources Research, Chinese Academy of Sciences, Beijing 100101, China; zhangxq@igsnrr.ac.cn
3   College of Geography and Environment, Shandong Normal University, Jinan 250358, China; wangyuzhe@sdnu.edu.cn
4   Earth and Environmental Sciences Programme, Faculty of Science, The Chinese University of Hong Kong, Hong Kong 999077, China; liulin@cuhk.edu.hk
5   Key Laboratory of Remote Sensing of Gansu Province, Heihe Remote Sensing Experimental Research Station, Northwest Institute of Eco-Environment and Resources, Chinese Academy of Sciences, Lanzhou 730000, China; dailiyun@lzb.ac.cn (L.D.); ranyh@lzb.ac.cn (Y.R.); chetao@lzb.ac.cn (T.C.)
6   Key Lab of Microwave Remote Sensing, National Space Science Center, Chinese Academy of Sciences, Beijing 100190, China; zhailimin21@mails.ucas.ac.cn
7   University of Chinese Academy of Sciences, Beijing 100049, China
8   Aerospace Information Research Institute, Chinese Academy of Sciences, Beijing 100094, China; qiuyb@aircas.ac.cn (Y.Q.); shilj@radi.ac.cn (L.S.)
9   College of Urban and Environmental Sciences, Northwest University, Xi'an 710127, China; nlwang@lzb.ac.cn
*   Correspondence: lpzhu@itpcas.ac.cn; Tel.: +86-10-84097093

**Abstract:** Over the past decades, the cryosphere has changed significantly in High Mountain Asia (HMA), leading to multiple natural hazards such as rock–ice avalanches, glacier collapse, debris flows, landslides, and glacial lake outburst floods (GLOFs). Monitoring cryosphere change and evaluating its hydrological effects are essential for studying climate change, the hydrological cycle, water resource management, and natural disaster mitigation and prevention. However, knowledge gaps, data uncertainties, and other substantial challenges limit comprehensive research in climate–cryosphere–hydrology–hazard systems. To address this, we provide an up-to-date, comprehensive, multidisciplinary review of remote sensing techniques in cryosphere studies, demonstrating primary methodologies for delineating glaciers and measuring geodetic glacier mass balance change, glacier thickness, glacier motion or ice velocity, snow extent and water equivalent, frozen ground or frozen soil, lake ice, and glacier-related hazards. The principal results and data achievements are summarized, including URL links for available products and related data platforms. We then describe the main challenges for cryosphere monitoring using satellite-based datasets. Among these challenges, the most significant limitations in accurate data inversion from remotely sensed data are attributed to the high uncertainties and inconsistent estimations due to rough terrain, the various techniques employed, data variability across the same regions (e.g., glacier mass balance change, snow depth retrieval, and the active layer thickness of frozen ground), and poor-quality optical images due to cloudy weather. The paucity of ground observations and validations with few long-term, continuous datasets also limits the utilization of satellite-based cryosphere studies and large-scale hydrological models. Lastly, we address potential breakthroughs in future studies, i.e., (1) outlining debris-covered glacier margins explicitly involving glacier areas in rough mountain shadows, (2) developing highly accurate snow depth retrieval methods by establishing a microwave emission model of snowpack in mountainous regions, (3) advancing techniques for subsurface complex freeze–thaw process observations from space, (4) filling knowledge gaps on scattering mechanisms varying with surface features (e.g., lake ice thickness and varying snow features on lake ice), and (5) improving

and cross-verifying the data retrieval accuracy by combining different remote sensing techniques and physical models using machine learning methods and assimilation of multiple high-temporal-resolution datasets from multiple platforms. This comprehensive, multidisciplinary review highlights cryospheric studies incorporating spaceborne observations and hydrological models from diversified techniques/methodologies (e.g., multi-spectral optical data with thermal bands, SAR, InSAR, passive microwave, and altimetry), providing a valuable reference for what scientists have achieved in cryosphere change research and its hydrological effects on the Third Pole.

**Keywords:** glacier inventory; snow depth; frozen soil surface freeze–thaw state; rock glacier; Aufeis; lake ice thickness; hazard; disaster; hydrology process; water cycle; Asian Water Tower; the Third Pole; the Tibetan Plateau (TP); the Qinghai–Tibet Plateau (QTP); the High Mountain Asia (HMA)

## 1. Introduction

The term "cryosphere" refers to Earth's frozen regions and includes glaciers, snow, river and lake ice, sea ice, ice shelves and ice sheets, and frozen ground. The cryosphere in High Mountain Asia (HMA) holds the most extensive global reservoir of solid water outside the polar regions. It is identified as the Asian Water Tower, feeding multiple large rivers and lakes that provide water for nearly two billion people [1]. The cryosphere in HMA is rapidly changing due to global warming. Glaciers are melting, permafrost is thawing, snow cover is decreasing, and the existence periods of lake ice are shortening, significantly impacting regional water resources, ecosystems, and human livelihoods. As HMA's cryosphere changes [2,3], cryospheric-related disasters have occurred, such as rock–ice avalanches [4,5], glacier collapses or detachments [6,7], debris flow [8], landslides [9], and glacial lake outburst floods (GLOFs) [10,11]. Historical data demonstrate that there were 697 individual GLOFs that occurred in HMA from 1833 to 2022, resulting in 6906 fatalities [12]. More recent data indicate that there were 681 snow and ice avalanche events that occurred in HMA between 1972 and 2022, resulting in more than 3100 fatalities [13].

HMA is the only region where glaciers with mass gain have been observed in Karakoram and Pamir [14] and Western Kunlun Shan with slightly positive mass balances [15], which is termed the "Karakoram Anomaly". These regions are characterized by a high concentration of glaciers with unstable flow dynamics, including glacier surging [16,17]. However, glacier change mechanisms in these regions are poorly understood due to the absence of glaciological and meteorological measurements [18], which may lead to high uncertainties in estimating potential GLOFs or other glacier-related disasters. Hence, monitoring cryosphere changes in a warming climate and obtaining datasets for subsequent analysis is crucial, given the observed impacts on hazards, water resources, hydrological cycling, global climatic change, and even sea-level rise [19].

The HMA cryosphere has been extensively studied, with more than eleven thousand papers utilizing satellite-based analyses published on the Web of Science and its extended library through 1 February 2024. Among these papers, glaciers are the most studied topic with 3435 published articles, followed by snow research with 2269 articles, and hazards with 829 publications (Figure 1). In addition, glacier studies have been cited more than thirty thousand times in the Web of Science core library (Figure 1). Both publications and citations of glacier articles doubled every five years between 2000 and 2015. Our extensive literature review provides a foundational overview of cryospheric studies, highlights research efforts incorporating spaceborne observations and cryo-hydrological modeling from diverse techniques/methodologies, and provides a valuable reference for glacier-related hazards and hydrological effects in HMA.

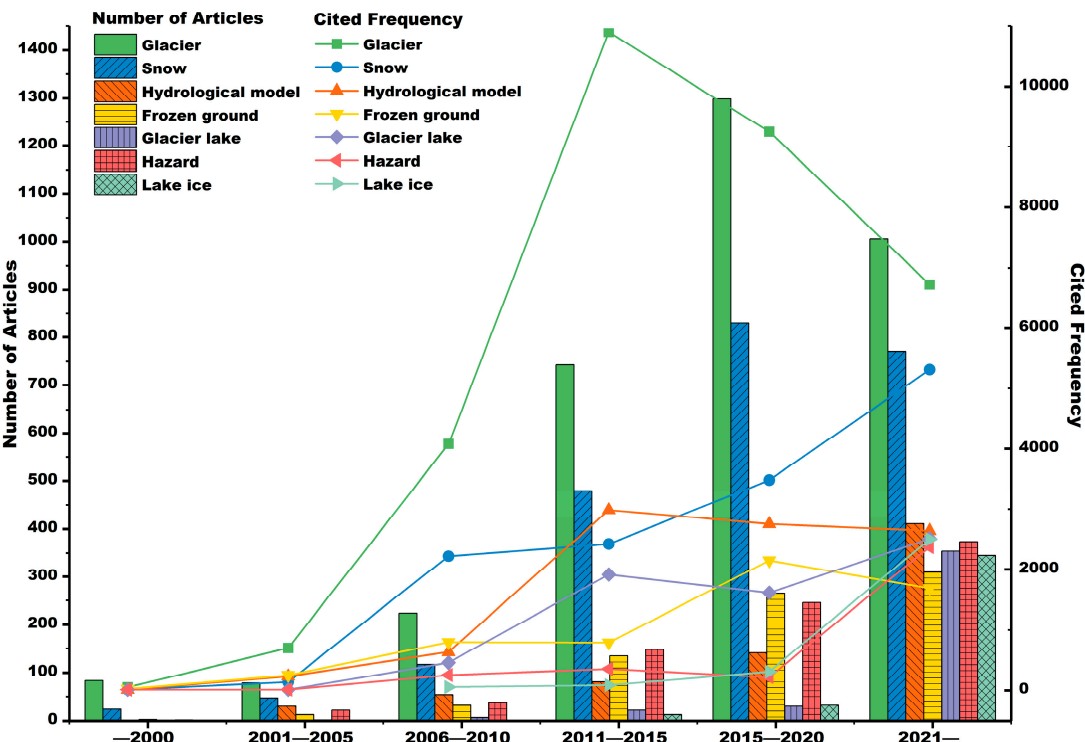

**Figure 1.** Literature on spaceborne cryosphere studies and hydrological models in HMA.

To structure the contents more systematically, an overview table, Table 1, lists all tables in the text and Supplementary Materials with a short description of their contents. As a vast amount of satellite-based data products were developed to study the cryosphere, an overview table sorted for these products is presented in Supplementary Table S1.

**Table 1.** An index of all tables.

| Table ID | Contents in the Table | Section |
|---|---|---|
| Table 1 | An index of all tables | Section 1 |
| Table 2 | Major glacier coverage data and inventories in HMA | Section 3.1 |
| Table 3 | Major reports of glacier mass change in HMA in different studies | Section 3.2 |
| Table S1 | An overview table sorted for the products in the cryosphere from space | Sections 1 and 2.6 |
| Table S2 | Major satellites/sensors for cryosphere monitoring | Sections 2.1 and 5.1 |
| Table S3 | Satellite-based products of DEMs, surface elevation, or surface elevation difference (DH) | Sections 2.1, 2.2.2 and 4.1.2 |
| Table S4 | Methods for glacier studies from space | Sections 2.2.1 and 2.6 |
| Table S5 | Major surge-type glacier inventories in HMA | Section 2.2.4 |
| Table S6 | Major methods for snow and frozen ground studies from space | Sections 2.3 and 2.4 |
| Table S7 | Products of snow cover and frozen ground from space | Sections 2.3, 3.3 and 3.4 |
| Table S8 | Methods for lake ice studies from space | Section 2.5 |
| Table S9 | Results on geodetic glacier mass balance changes in HMA | Section 3.2 |
| Table S10 | Products of lake ice studies from space | Section 3.5 |
| Table S11 | Download linkage for datasets, databases, or platforms | Section 3.6 |
| Table S12 | Acronyms used in the paper | All |

## 2. Datasets and Methodologies in Cryosphere Monitoring

Cryospheric studies utilizing in situ and remote methodologies deepen the ever-increasing understanding of the global implications of changes occurring in the cryosphere. However, in situ observations are costly and time-consuming, thus covering only limited geographical areas. Complimentary to this, studying the cryosphere with satellite-based technology overcomes in situ limitations and provides robust, large-scale geographic coverage with comprehensive, long-term monitoring capabilities. Therefore, remote sensing

technology and instruments aboard Earth-observing satellites, which ushered in a new era of cryosphere exploration in the 1950s, make measurements with incredible precision possible for late-20th-century researchers.

### 2.1. Input Datasets from Space

More than fifty satellites with remote sensors to monitor the cryosphere have been widely deployed since the 1950s. Current and evolving satellite-based instruments and platforms to study the cryosphere include optical sensors such as multi-spectral, hyperspectral, and Light Detection and Ranging (LiDAR); microwave techniques involving passive and active microwave sensors, e.g., Radio Detection and Ranging (RADAR), and Interferometric Synthetic Aperture Radar (InSAR); and various altimetry techniques involving Lidar and SAR altimeter (Figure 2). In addition, rapid technological advancements have made satellite images with higher spatial resolution and fewer revisiting days more available than ever (Figure 2). Supplementary Table S2 summarizes the most frequently used satellites/sensors for cryosphere observations.

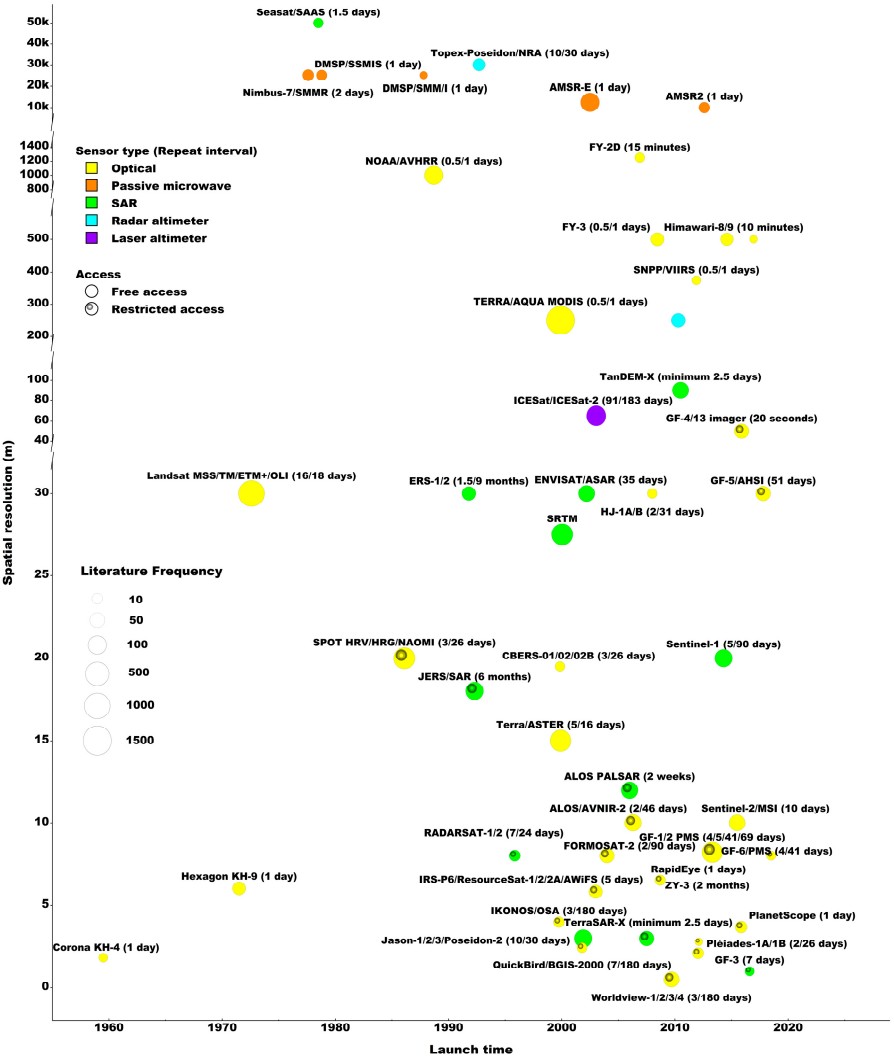

**Figure 2.** Frequency of occurrence in the literature on spaceborne sensors for cryosphere monitoring.

Literature analysis demonstrates that the Landsat image series dating back to the 1970s is the most frequently used high-resolution data to study the cryosphere over the past five decades (Figure 2). It involves images from the Multispectral Scanner (MSS) of Landsat 1–5 since 1972, the Thermal Infrared Sensor (TM) onboard Landsat 4 and 5 from the 1980s to 2013, the Enhanced Thematic Mapper Plus (ETM+) of Landsat 7 since

1999, Landsat 8/Operational Land Imager (OLI) since 2013, and the Landsat 9/OLI-2 since 27 September 2021. Access to all Landsat images has been free of charge since 2008, and data availability in Google Earth Engine and similar platforms/tools have significantly improved data processing efficiency from the 2010s. Similarly, the Sentinel series satellites, e.g., Sentinel-1/SAR-C and Sentinel-2/MSI (Multi-Spectral Imager), were deployed in 2015 and have been used extensively across various fields of Earth observation due to their high spatiotemporal resolution with global coverage in 5 days for the "Extra-wide swath" mode on Sentinel-1, and a 10-day revisit time of Sentinel-2.

In addition, the Moderate Resolution Imaging Spectroradiometer (MODIS), the key instrument aboard the Terra and Aqua, has been the most frequently used medium-resolution spaceborne data for cryosphere studies over the past two decades (Figure 2, Table S2). Due to its daily twice-revisit interval, MODIS is broadly applied in cryospheric studies, e.g., snow, land surface temperature (LST), and lake ice cover.

The Advanced Spaceborne Thermal Emission and Reflection Radiometer (ASTER), the only high spatial resolution instrument aboard the Terra, has been broadly applied in cryosphere studies due to its 14-channel radiometer across three sub-systems, involving Visible/Near-Infrared (VNIR), Shortwave Infrared (SWIR), and Thermal Infrared (TIR). The high-resolution multi-spectral VIS/NIR imager has not only been used for glacier mapping [20] but also for generating a Digital Elevation Model (DEM) globally with changing viewing angles, which provides optical stereo pair images, i.e., the ASTER GDEM V1 [21], or ASTER GDEM V2/V3 (Table S3).

Additional high spatiotemporal resolution optical sensors have been operated since 2000, e.g., High-Resolution Geometrical (HRG) on SPOT 5, Three-line Array Camera (TAC) on Zi Yuan (ZY)-3, and SpaceView-110 Imaging System on Worldview-4. These technologies obtain images every four days by programming side-swing operations, e.g., Gao Fen (GF)-1/Panchromatic and Multi-spectral CCD Camera (PMS), GF-2/PMS-2, or GF-6/PMS (Table S2). However, the high price of these images limits their usage in cryosphere studies. Moreover, unfavorable meteorological conditions, e.g., cloud cover, rain, or fog, prevent optical sensors from obtaining high-quality images, complicating the application of these technologies and the data collection required to study sudden disasters in the cryosphere.

In contrast, microwave sensors present significant advantages in all weather conditions. Passive microwave radiometer, for example, the Scanning Multichannel Microwave Radiometer (SMMR) on Nimbus-7, provided global revisiting data every two days from 1978 to 1994. Similarly, Advanced Microwave Scanning Radiometer-2 (AMSR2) can cover the Earth daily with a 10 km pixel size since 2012. In contrast, active microwave sensors take advantage of high resolution and penetrating abilities into the land subsurface. Among these, Synthetic Aperture Radar (SAR) is the most frequently used active microwave technique. For instance, the Phased-Array L-band Synthetic Aperture Radar onboard Advanced Land Observing Satellite (ALOS/PALSAR) provided 7–100 m SAR data from 2006 to 2011. However, the long revisiting time interval of a minimum of two weeks limits its applications across rapidly changing observations. To date, Constellation Mission can provide a shortened revisit time of within a few days. For example, the RadarSat Constellation Mission, launched on 12 June 2019, allows for a repeat cycle of four days.

Despite these advances, not all methods are mature for estimating glacier, snow, permafrost, and lake ice phenology or properties. Therefore, an increasing number of remote sensing techniques, quantitative inversion methodologies, and abundant data products from models have been developed to estimate glacier mass balance (MB), glacier velocity/surging, glacier thickness, snow water equivalent (SWE), snowline/equilibrium line, snow depth, and albedo. While the current satellite-based image techniques appear sufficient for exploring glacier area, glacier length, and snow cover extent, high uncertainties in these parameters exist because of the diverse landforms and rough terrain and spectral similarities, especially for debris-covered glacier areas.

## 2.2. Glacier

### 2.2.1. Glacier Delineations

Satellite-based remote sensing techniques, specifically microwave and optical technologies, have been broadly applied in global- and regional-scale surveys of alpine glaciers due to the large number of glaciers and their relative remoteness [22,23]. By extension, the evolution and application of these technologies have been successfully adopted in HMA to measure glacier coverage, glacier area changes, glacier velocity/flow, glacier surging, and geodetic glacier mass balance. In addition, nearly all studies described were made possible via free access to satellite data and DEMs, to which we are thankful for and indebted to those space agencies and governmental organizations. We summarize the primary methodologies for glacier delineations in HMA in Supplementary Table S4.

Traditional techniques of glacier delineation are generally based on multi-spectral satellite images. The commonly used multi-spectral classification methods include traditional manual digitization [20,24–26] and semi-automatic and automatic algorithms for glacier mapping with satellite data. The band algebraic operation, namely, multi-spectral classification by threshold segmentation, is the core technique for the semi-automatic and automatic algorithms, including the spectral-band ratio [27–32], Normalized Difference Snow Index (NDSI) [33], and unsupervised or supervised classification techniques [20]. Compared with band-based segmentation by image pixels noted above, object-oriented methods accommodating adjacent pixels may limit the salt-and-pepper noise in higher-resolution images, connecting homogenous regions and improving accuracy and efficiency. However, the object-based image analysis (OBIA) configuration is quite complex, and the software is costly. By contrast, spatial filtering of the resulting glacier mask also works in many cases to reduce the salt-and-pepper noise. Furthermore, many more innovative automatic classification schemes are utilized to map glaciers with artificial intelligence (AI), e.g., machine learning classifiers (i.e., support vector machines, k-nearest neighbors, random forest, decision trees, gradient boosting, multi-layer perception, and artificial neural network methods) [34–36]. Nevertheless, greater innovation does not directly correlate to greater efficiency and validity. Applying machine or deep learning methodologies is significantly more complex, with a minimal gain of accuracy with an ongoing need for manual corrections.

Delineation of debris-covered glaciers continues to challenge ice identification because of the similar spectral properties in the optical images between debris-covered ice and its surrounding valley rocks [37] and/or the spectral similarity with a gradual transition to ice-cored moraines and other ice-debris landforms in outwash plains. Therefore, the debris-covered glaciers are outlined with more complex techniques, including semi-automatic methods using optical satellite data obtained by multi-spectral and thermal bands with topographic characteristics (e.g., slope) from DEM [38,39]. The thermal bands, particularly ASTER (B10–14, 8.125–11.65 μm) and Landsat ETM+ (B6, 10.4–12.5 μm), were adopted to distinguish debris cover from glaciers [40,41]. Such approaches, however, are generally unsuitable for mapping heavily laden debris-covered glaciers [40]. Given that manual glacier delineation is problematic for debris cover and/or snowfields [42], many fully automatic methods have emerged [43]. Additionally, the comprehensive multi-platform datasets of SAR coherence images were also combined and applied with visible, thermal spectrums and DEM [44].

As noted, machine/deep learning algorithms have been widely utilized to map debris-covered glaciers [45–49], including artificial neural networks (ANN), the support vector machine (SVM), the random forest (RF) classifier [34], and convolutional neural networks (CNN). Nevertheless, applying these machine learning techniques is limited by cloud cover, rough terrain, mountain shadows, and freshly frozen lakes [49]. Consequently, various neural network algorithms have been developed for delineating debris-covered glaciers, such as combining spatial, spectral, and geomorphometric features with DEM [47,50]. These machine learning classifiers can achieve high accuracy in detecting debris-covered ice based on the combined use of SAR coherence data, visible and thermal images, and

geomorphometric parameters from a DEM [51]. However, the accuracy decreased because of the limited number of available training samples from existing glacier inventories, small study areas, and difficulties in hyper-parameter optimization. In short, the recognition of debris-covered glaciers using these methods depends heavily on the debris surface/texture characteristics, which often fails to clearly explain that the detected debris areas belong to a glacier lying underneath the debris.

Although the on-screen glacier ice delineation is labor-intensive and time-consuming, manual digitization and visual inspection are still widely used to maximize the accuracy of results [24,52–54]. The data accuracy of debris-covered glaciers can be significantly improved by a manually intensive verification and correction effort based on the draft results from the band ratio NIR/SWIR [43], especially when the analyst is familiar with the study area and is experienced with high-resolution images.

Common methods concentrate on the individual glacier delineation results from multi-spectral images at different epochs obtained by different methodologies, showing the combined results with overlaid vector–vector or vector–raster data in a map [22]. However, these approaches ignore the conflicts or inconsistencies of individual image results, like noise from diversified sources that are difficult to distinguish [55]. The prominent noise can be attributed to the highly variable seasonal snow, differences in interpreting debris-covered regions, misclassification due to various principles or definitions of "glacier", regional differentiation in cast shadows or perennial snow cover, uncertainties in manual mapping, errors of co-registration and geolocation, and that multiple data sources from satellite images, aerial photography, topographic data, and available historical maps [56,57].

### 2.2.2. Glacier Mass Change

The satellite-based geodetic glacier mass balance (MB) is primarily derived from surface elevation differencing (Supplementary Tables S1 and S3, Figure 3), i.e., between DEMs generated from photogrammetric images [58–60] or radar imagery, the bistatic differential SAR interferometry (DInSAR) [61], and satellite altimetry measurements, e.g., ICESat/GLAS [62], ICESat-2/ATLAS [63,64], and CryoSat-2 [23,65]. Recently, an albedo-MB approach was presented to evaluate glacier-wide surface annual MB change for large debris-free glacier areas [66,67]. Moreover, gravity-based measurement is also applied to estimate large-scale groundwater storage change with solid water melt involving glaciers, permafrost, and lake mass change in HMA [68,69]. The satellite-based surface elevation measurements and datasets are summarized in Supplementary Table S3 and presented in Figure 3.

Limited by the elevation measuring uncertainty [70], the estimation accuracy of geodetic glacier mass change is also closely related to the accuracy of DEMs for calculating glacier surface differences. However, DEMs from optical stereo pairs have gap areas in shadows of steep terrain and low contrast in the firn zones, leading to the lower quality of DEMs in rough mountainous terrains [21,71]. Bistatic SAR interferometry (InSAR) (i.e., SRTM and the TanDEM-X/TerraSAR-X) launched a new era of satellite-based radar. This remote sensing technique generates a consistent global DEM with unprecedented accuracy. However, SAR's technology of employing side-looking geometry leads to geometric distortion (e.g., layover, foreshortening, and shadows with gap areas). In addition, substantial penetration uncertainties remain due to the radar signals penetrating snow and ice. As such, there are also high uncertainties in alpine elevations when using the SRTM DEM. In comparison, the NASA DEM, the updated version of the SRTM DEM, has higher accuracy. Nevertheless, considerable uncertainties remain when utilizing InSAR-based DEMs to estimate glacier geodetic MB changes, even with the corrected penetration depth into snow and ice.

In contrast, laser altimetry could provide sufficient precision for surface elevation changes in glaciers, lakes, or frozen ground by averaging hundreds of footprints at different intervals. The greater the number of footprints averaged across a given region, the greater the precision of the surface elevation investigations. Therefore, the laser altimetry measurements by ICESat/GLAS operated from 2003 to 2008 and ICESat-2/ATLAS since

September 2018 offer precise surface height observations (the accuracy is approximately 2.54 cm). For example, Fan et al. (2022) estimated the glacier MB in HMA from 2000 to 2021 with the NASADEM and ICESat-2/ATLAS data [64]. These footprints, however, do not cover each glacier's surface, especially the smaller ones. DEMs derived from optical stereo pairs and InSAR remain irreplaceable because of their broadly continuous spatial coverage, specifically for the vast number of small glaciers in HMA. As reviewed, the primary free accessible global DEMs include the SRTM DEM, TanDEM-X DEM, the NASA DEM, ALOS World 3D DEM, the Copernicus DEM with GLO-90 at 90 m and GLO-30 at 30 m resolution, the TanDEM-X 30 m Edited Digital Elevation Model (EDEM), and the HMA_DEM8m_MOS (Supplementary Table S3; Figure S1).

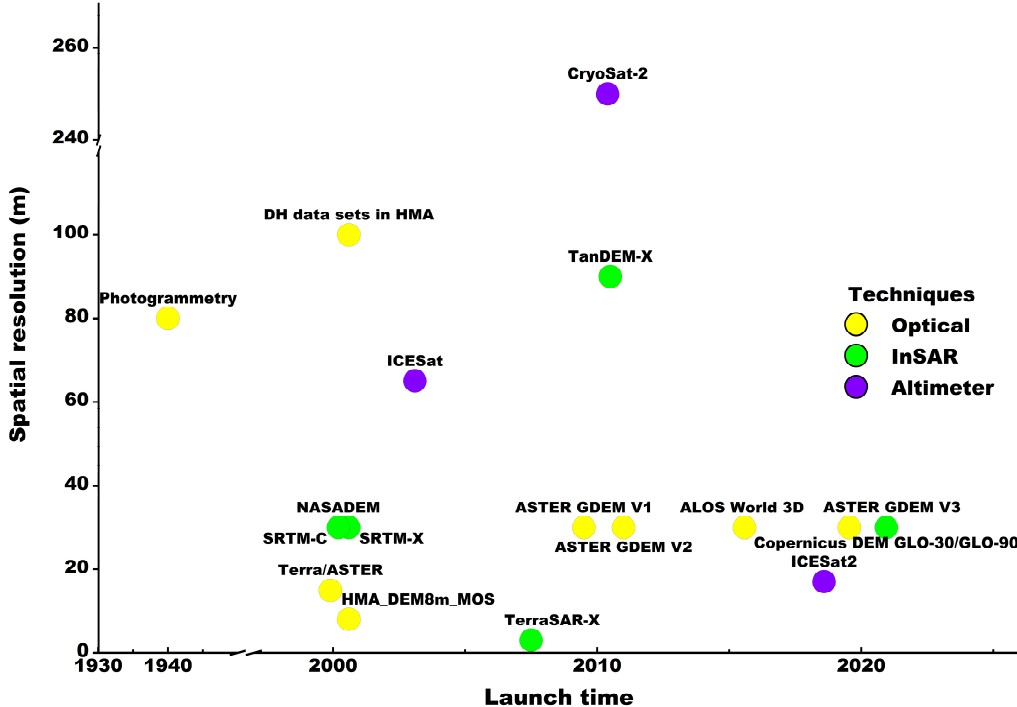

**Figure 3.** Available DEMs, surface elevation, or surface elevation difference (DH) data in HMA.

Over the past five years, high-spatial-resolution DEMs have been quantitatively created from optical stereo photos over the glacierized areas in HMA. For example, more than 50,000 DEMs were created from ASTER L1A images over HMA between 2000 and 2016, using enhanced and fully automated methodologies [72]. In addition, approximately half a million DEMs at 30 m spatial resolution were produced to estimate the global glacier surface elevation changes with specific statistical methods and modern photogrammetry techniques, covering more than 20 times the Earth's land area, based on the openly available NASA's ASTER archives and airborne elevation datasets between 2000 and 2020 [73]. Moreover, an estimated 99% of HMA glaciers were covered with 28,278 ASTER DEMs between 2000 and 2018 [59] and 5797 high-resolution DEMs from sub-meter level commercial optical stereo imagery (DigitalGlobe WorldView-1/2/3 and GeoEye-1) between 2007 and 2018, primarily between 2013 and 2017. The sub-meter resolution of stereo pairs derived fine-scale terrain features with sufficient contrast in the images and almost no saturation gaps. Further, the higher the spatial resolution of the stereo-pairs, the higher the vertical accuracy of the generated DEMs [23].

### 2.2.3. Glacier Velocity

Glacier surface velocities can be derived from satellite-based optical data [74–76], SAR images [77], or from both optical and SAR images [19,78]. The field-based measurements of ice flow, or ice velocity, relied on the in situ stakes drilled into glacier ice with positioning



measurements by GPS or DGPS. Along with the wide application of satellite observations in glacier surface velocities at large scales, innovative techniques have been broadly applied to derive ice velocities, such as feature tracking based on optical imagery [76,79], InSAR [80,81], and SAR offset-tracking [77,82].

Among these glacier velocity products, two projects generated datasets derived from optical images, namely, the NASA MEaSUREs ITS_LIVE project and the NASA Global Land Ice Velocity Extraction (GoLIVE) project. The former provided image-pair velocities at 240 m resolution based on Landsat satellite imagery since 1985 [76], processing annual data from optical, radar, and laser satellite sensors to update products of glacier surface elevation change and surface velocities. The latter has been estimating mountain glacier motion and ice velocity since May 2013, using a 300 m pixel size with a 16-day temporal resolution [74,75]. However, ITS_LIVE and GoLIVE velocity data are limited by optical image acquisition capacity, quality, and spatial resolution. Additionally, ITS_LIVE does not cover land ice areas smaller than 5 km$^2$. Also, the near-global glacier velocity product of Sentinel-1 was released in 2014 and continuously updated, offering data at 200 m resolution per month or an annual average [77]. Furthermore, the synthesized ice motion products with a 50 m resolution are released based on multi-sensors from both optical and SAR image pairs involving Sentinel-2A/B, Landsat 8, and Sentinel-1A/B, and the model outputs, covering 98% of glacierized areas on Earth from 2017 to 2018, upgrading its applicability and resolution [19].

### 2.2.4. Glacier Surging

Surge-type glaciers are essential to understanding glacier flow instability, which increases potential hazards by blocking downstream rivers, inundating the proglacial areas, and facilitating GLOFs [83–85].

Glacier surging has been differentiated using the "surge index" on a global scale [86]. These surge indices involve a sudden and significant glacier terminus advance, an increase in flow velocities at the active phase which may reach at least an order of magnitude higher than the usual quiescent status [87,88], surface elevation change (e.g., terminus steepening and thickening [17], downwasting in the reservoir/accumulation zone [89], and a combination of geomorphological evidence of oscillations in flow (e.g., bulged terminus, heavy surface crevassing, looped moraines, eskers, push moraines, etc.) [90]. In distinguishing surging glaciers from advancing glaciers, it was noted that the surging glaciers experienced a dramatic fluctuation in surface velocities during and after the surge event. In contrast, the advancing glaciers flowed in a stable mode [91].

According to the surge indices, glaciers surging have been identified by the methodologies of visual interpretation, DEM differencing, and surface velocities based on satellite data in HMA. Notably, manual digitization is always applied to outline the multi-temporal surge-related glacier terminus position. For example, glacier surge-affected areas were differentiated manually, corroborated by very-high-resolution optical images from Bing Maps and Google Earth alongside multi-temporal changes in glacier velocity and surface elevation throughout the HMA [89]. In another example, surge-indicative featured glaciers in HMA were manually identified in Google Earth, and their terminus change was digitized utilizing the Google Earth Engine Digitisation Tool (GEEDiT, [92]) based on multi-temporal remotely sensed imagery [84]. In addition, surge-indicative morphological changes could be visually identified based on the animated image merging from multiple time series images for recognized surging glaciers, which were characterized by evident lower thickening and upper thinning from glacier surface elevation differences between DEMs at different epochs [16,93]. This demonstrates that the traditional man-machine interaction remained indispensable, especially visual interpretation and manual digitization based on optical images. Therefore, studies on glacier surging in HMA remain challenging related to their irregular behavior and our inadequate understanding of their thermal and hydrological conditions and impacts. Present analytical methods remain insufficient with limitations of very-high-spatiotemporal-resolution surface elevation data and images from space.

Currently, there are several inventories of surging glaciers in HMA [16,84,86,89,94], which presents a significant difference in the number of identified surge-type glaciers varying from 137 to 890 pieces (Table S5). In addition, the most frequent occurrence of glacier surging was found within geographical clusters, where surge-type glacier inventories have been studied in Karakoram [17,64,90,95–98], Pamir [93,99], Tian/Tien Shan [100], and Kunlun Mountains [101,102].

Glacier surge has also been compiled in the Tibetan Interior Mountains, Mt. Himalayas, Tanggula, Qilian, Nyainqêntanglha, Pamir Alay, Altun, and Gangdise regions (Table S5).

*2.3. Snow*

Satellite-based techniques are broadly applied to estimate snow cover extent, snow water equivalent (SWE), and snow depth, which are essential parameters in climatic–hydrological systems [103].

Snow cover extent is studied by a variety of remote sensing techniques [104–106], with optical remote sensing being the most common (Table S6). Snow cover can be efficiently outlined at high spatial resolution by NDSI (Normalized Difference Snow Index) based on spectral differences. However, the NDSI index does not distinguish between snow and (clean) glacier ice. NDSI thresholds for identifying snow cover change with regions, sensors, and weather conditions. For example, cloud cover decreases the temporal resolution of snow cover data. Therefore, most studies include a focus on developing local snow-cover algorithms for different sensors [107,108] and cloud-removing methods for snow-cover products [109]. Although snow-cover extent mapping methods are generally more mature than those used to map snow depth, steep high mountain terrain remains challenging in regions with significant spatial differences in snow-cover extent during melting periods.

Snow depth can be retrieved by passive microwave (e.g., AMSR-E, AMSR2) or active microwave (e.g., InSAR, SAR, and Lidar) from space [110]. The passive microwave technique is the most efficient in estimating SWE and snow depth in large areas due to its high temporal frequency over long periods. However, it has a very coarse spatial resolution. In addition, Chang et al. developed a brightness temperature gradient method to efficiently derive snow depth and SWE [111].

Considering the regional differentiation of snow properties and land cover, many studies develop location-specific algorithms that assimilate local snow characteristics [112]. Snow process models have improved the measurement of snow features, including the development of a physical snowpack model (SNOWPACK) [113] and the Snow Thermal Model (SNTHERM) [114,115]. In addition, the continuity characteristics of snow cover can be obtained by land surface models using the Snow Microwave Radiative Transfer model (SMRT) to improve the accuracy of snow depth estimations [116]. However, limitations remain, e.g., the difficulty of observing and measuring snow features at high altitudes or on steep mountain terrains. In parallel, satellite-based passive microwaves cannot describe the high spatial heterogeneity of snow cover in HMA despite its accuracy in measuring snow depth. Thus, previous studies developed downscaling methods to derive snow depth by combining multi-source data [109,117–120]. The accuracy of these downscaling methods is limited by the snow depth inversion of the passive microwave technique. Moreover, snow depth is severely underestimated in deep-snow areas because of the signal saturation of microwave radiation from snow [121].

As for active microwave techniques, SAR from space has been applied to detect snow depth since 2000. The present satellite-based SARs mainly utilize C and X frequencies, which intensely penetrate snowpack and are weakly scattered by snow grains. In sharp contrast, the change in soil dielectric properties leads to significant uncertainties for snow depth detection. In view of this, satellite Lidar has been applied since the late 2000s as an efficient method to detect snow depth at a high spatial resolution. However, satellite Lidar usage for fast-changing snow surface and depth is limited due to the long revisit time and cloud cover. Hence, airborne Lidar is currently the best technique to detect snow depth at a watershed basin scale. Nevertheless, the complicated terrain impacts the signal

and polarization direction, resulting in spatially different uncertainties in snow depth estimation in mountainous areas.

Measurement of snow products (Supplementary Table S7) with higher spatiotemporal resolution and accuracy is urgent for mountainous areas. Future snow research should be strengthened by combining obtained in situ investigation data and quantitative remote sensing data with developed machine learning technologies [122,123].

### 2.4. Frozen Ground

2.4.1. Frozen Soil

Frozen ground, also known as frozen soil, is primarily defined by the duration of freeze–thaw periods, which are classified into seasonally frozen ground and permafrost. Permafrost is ground at or below 0 °C for at least two consecutive years, while seasonally frozen ground freezes and thaws annually. Ground freezing and thawing generally occur in the active layers of seasonally frozen ground, short-term frozen ground, alternate-year frozen ground, and perennially frozen ground [124]. Recent advances in remote sensing techniques to study ground thawing and freezing offer significant advantages compared to earlier field-based and modeling works.

The thermal status of permafrost can be effectively obtained by the land surface temperature (LST) derived from optical, infrared remote sensing technology (e.g., ASTER, MODIS), characterized by long time series, high spectrum, and high resolution. Based on the LST products, it is possible to analyze the thermal stability of permafrost [125]. The permafrost's mean annual ground temperature (MAGT) can be obtained by statistical learning modeling with an extensible strategy based on satellite data. Figure 4 offers one example of estimated MAGT from 2000 to 2016, where permafrost is highly concentrated in the northern and the western TP, diminishing in the southern and the southeastern TP [126]. Nevertheless, visible light remote sensing technology could not observe parameters such as the surface freeze–thaw state, deformation, or active layer thickness. In contrast, microwave remote sensing technology exhibits significant advantages in all-weather and specific penetration capabilities. Therefore, integrating the optical and microwave techniques can effectively improve the temporal revisiting frequency and the spatial resolution for comprehensive observation, which can monitor the distribution and changes in frozen grounds over the long term.

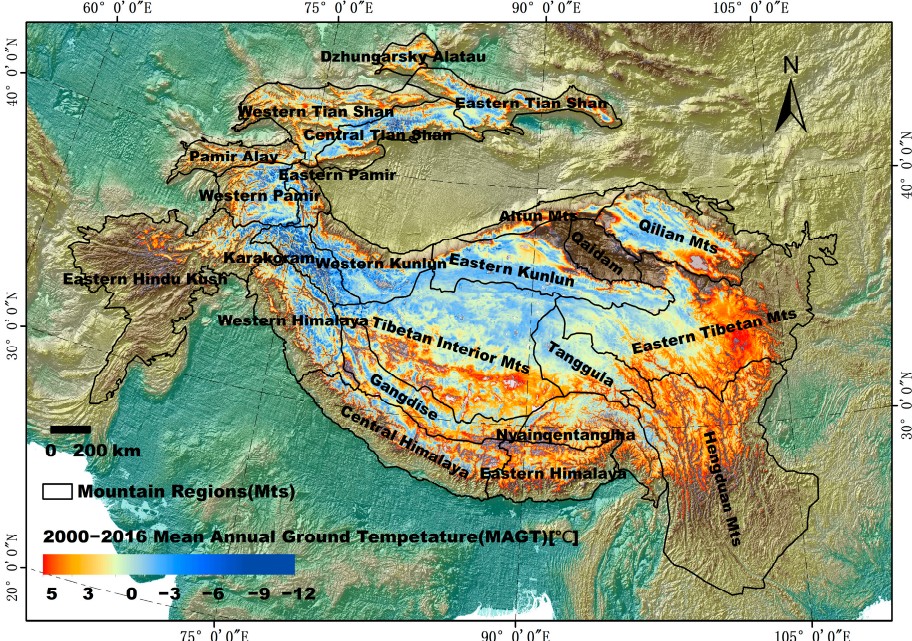

**Figure 4.** The mean annual ground temperature (MAGT) [126,127]. (The boundary of HMA is composed of the results by Zhang [128], Lu [29], and Shean [59]).

Passive microwave remote sensing is very sensitive to the dielectric changes in the freeze or thaw states of the ground, e.g., FY-3/MWRI [129]. Therefore, combining algorithms is necessary to improve the spatial resolution of passive microwave remote sensing data retrieval. Researchers could couple a series of algorithms designed for microwave radiometers (i.e., dual index, decision tree, freezing and thawing discriminant, seasonal threshold, and L-band relative freezing factor threshold) to achieve the surface freeze–thaw states products.

Active microwave remote sensing technology performs well in land surface deformation and the inversion of active layer thickness. We present the multi-source SAR techniques for monitoring frozen ground deformation (Supplementary Tables S1 and S6). For instance, InSAR is applied to monitor frozen ground deformation and temporal changes [130]. Additionally, D-InSAR technology is developed from InSAR to obtain data for slight deformations of frozen soil, analyze its spatiotemporal deformation characteristics, and monitor changes in the active layer of frozen ground. In addition, the Permanent Scatter Synthetic Aperture Radar Interferometry (PS-InSAR) and Small Baseline Subset Synthetic Aperture Radar Interferometry (SBAS-InSAR) technologies enable the monitoring of the ground surface deformation over vast permafrost areas [131].

Combining passive and active microwave remote sensing technology is more conducive to frozen ground research. However, the accuracy and the spatial resolution are low in steep, high-mountain terrain areas, where the inversion depth of frozen ground is shallow, and the long-term observations require verification. Furthermore, as the subsurface temperature is difficult to measure remotely, there are only a few satellite-based studies to quantify permafrost thermal state and creeping motion and to evaluate its hydrological effects in HMA [132].

To date, there are two typical modeling methods to estimate the ground temperature of frozen soil using remote sensing. One is to use LST as an input to an empirical frozen soil model, such as TTOP (the Temperature at the Top Of the Permafrost). The second is to establish a distribution model of frozen ground based on remote sensing, climate reanalysis, land cover, and other related data, which is an effective method for defining frozen ground areas with insufficient in situ observational data. Notably, the estimations from these distribution models are consistent with the actual borehole data despite the influence of regional heterogeneity, guaranteeing relatively high accuracy.

### 2.4.2. Rock Glaciers

Rock glacier is a unique type of periglacial landform that moves due to the gravity-driven creeping of permafrost. Further, intact rock glaciers also contain significant ground ice and release fresh water in summer [133]. Therefore, measurement of rock glaciers' water storage capabilities is an important component of HMA hydrology [132].

The number and areas of rock glaciers have been investigated by multiple spaceborne data, such as optical images and InSAR techniques (Table S6). The first near-global rock glacier database using spaceborne data was achieved by meta-analysis of 131 inventory studies and publications produced prior to October 2017 [134]. However, inherent flaws in the meta-analysis methodology likely led to large overall uncertainty. Moreover, there are few inventories of rock glaciers on the TP. In Daxue Shan, southeastern Tibet, rock glaciers in the inventory were manually digitized based on high-resolution satellite imagery from Google Earth [135]. A similar method was applied to obtain a rock glacier inventory in Guokalariju on the southeastern TP [136]. In the Nepalese Himalaya, a rock glacier inventory was compiled according to freely available SPOT and DigitalGlobe (e.g., QuickBird, IKONOS, Worldview-1 and 2) via the Google Earth platform [137]. In the Karakorum, the northern Tien Shan, and the Altai, a rock glacier inventory was established using high-resolution images over a digital globe, Google Earth (GeoEye, SPOT 5, Quick-Bird, WorldView, IKONOS), SRTM, and WorldClim data [138]. In Northern Tian/Tien Shan, active rock glaciers' creep speeds were observed by DInSAR techniques, including ALOS-1 PALSAR-1, ALOS-2 PALSAR-2, ERS-1/2, TanDEM-X DEM, Sentinel-1, and Google

Earth/Map and Bing Maps [139]. In the western Nyainqêntanglha mountain, an inventory of active rock glaciers was derived by a combination of manual outlining and automatic classification according to InSAR techniques by Sentinel-1 in 2016–2019, TanDEM-X DEM, optical data from Landsat images from Bing Maps, Zoom Earth, and Google Earth, Sentinel-2 [133]. Similarly, in the Himalaya rock glacier inventory, fine-resolution CNES/Airbus (e.g., Pléiades and SPOT), DigitalGlobe-derived images (e.g., QuickBird, Worldview-1 and 2), and SRTM DEM were used within the Google Earth Pro platform [132]. In addition to the methodology of semi-automatic classification or manual interpretation, recent attempts in deep learning classifiers were applied to automate rock glacier mapping using very-high-resolution optical and SAR spaceborne images [140–142].

### 2.4.3. Aufeis

Aufeis (or icing) is "a sheet-like mass of layered ice formed on the ground surface, or on river or lake ice, by freezing of successive flows of water that may seep from the ground, flow from a spring or emerge from below river ice through fractures" [143]. Aufeis are widely distributed in cold regions, especially in permafrost areas. They act as a temporary store of water in winter and discharge water in early spring, playing an essential role in permafrost hydrology. The formation of massive Aufeis may cause hazards by destroying buildings and covering roads.

Aufeis in HMA has not been extensively studied, and only a few attempts have been made to map their spatial distribution in select regions. Brombierstäudl et al. (2021) applied time-series analysis to Landsat-5 TM/7 ETM+/8 OLI images and used the seasonal variations (amplitude and phase) of NDSI to detect large Aufeis fields in the Upper Indus Basin [144]. The authors identified 3848 recurrent Aufeis, which covered an area of 298 km$^2$. Using a similar method and Landsat-8 OLI thermal infrared bands, Gagarin et al. (2022) identified 1659 Aufeis across the Kunlun Mountains [145]. Twenty-seven recurrent Aufeis fields were identified by using a random forest classifier based on Sentinel-2 and Landsat-5 TM/8 OLI images in the Tso Moriri basin in India [146]. These initial results raise more interest in studying Aufeis and their hydrological significance in HMA.

Major methods for frozen ground studies from space are summarized in Supplementary Table S6.

### 2.5. Lake Ice

Lake ice is a crucial indicator of global warming [147]. Lake ice phenology (i.e., lake ice appearance, formation, area, thickness, and existence duration) is affected by temperature variation [148], and is essential to lake thermodynamics. Furthermore, lake ice freezing and melting may significantly affect the aquatic ecosystem and change its biogeochemical processes, e.g., influencing plankton growth and causing anoxia in deep water [149].

The research on lake ice changes is based on ground observations, satellite remote sensing, and modeling [150,151]. Unfortunately, ground-based lake ice observations on the TP are inadequate. Therefore, satellite-based observations, including optical, passive, and active microwave data, are broadly applied in studying lake ice phenology [152]. Specifically, research on lake ice phenology often utilizes band ratio threshold methods based on optical remote sensing images (e.g., Landsat [153], AVHRR [154], or MODIS [155]). In particular, high-spectral MODIS data are widely used for its daily revisiting interval on a global scale. However, lake ice data are not available when lakes are covered by clouds or if deep shadows are present in the optical images. Further, brightness temperature from passive microwave remote sensors (e.g., SMMR [156–158], AMSR-E [159], and SSM/I [160]) is performed to retrieve lake ice phenology and thickness with low spatial resolution. Finally, the active microwave technique is proficient in monitoring lake ice phenology with sensors such as ERS-1/2 SAR [161] and Radarsat-1/2 SAR [162], although its narrow swath width and relatively low temporal resolution limit its application. Lastly, a satellite-based radar altimeter has been applied to retrieve lake ice thickness [158].

Primary methods for lake ice studies from space are summarized in Supplementary Table S8.

*2.6. Glacier-Related Hazards*

The changing cryosphere in HMA has led to instability of paraglacial, glacial and periglacial circumstances resulting in various hazards [163], involving snow/ice/rock/debris avalanches, glacier detachments or ice collapse, glacier surges, snowstorms, landslides, debris/ice flows, floods generated by glacial lake outbursts, thermokarst development, and instability and subsidence of building foundations due to degradation of permafrost [164].

Satellite images are widely applied in observing cryospheric hazards (Table S4), whereof only a few glacier detachments or avalanches have been recorded to date. In 2015, a glacier avalanche on Mount Tobe Feng in the southeastern Pamir which engulfed 1000 ha of pasture, hundreds of livestock, and 61 dwellings was observed using Landsat 8 images and GDEM [165]. In 2016, the collapse of the two adjacent Aru glaciers was visually studied using Corona, Landsat, high-resolution optical and radar images, DEMs from SRTM, TanDEM-X, and ASTER, ICESat laser altimetry, coupled with mass-balance modeling [166]. In the same year, a glacier collapsed in the Amney Machen mountain was captured by a time series of freely available images involving Corona KH-4, Landsat, Sentinel 2, very-high-resolution imagery in Google Earth and maps from earthexplorer.usgs.gov, which found repeat surges and collapses in 2004, 2007, and 2016 [167]. In 2018, two glacier collapse-induced river blocks in the Yarlung Tsangpo were investigated using Sentinel-1, 2, GaoFen-1 (or GF-1), and PlanetScope images, along with images from unmanned aerial vehicles and helicopters [6]. In 2019, the Shispare glacier surging in 2017–2019, was investigated by Landsat, ASTER, Sentinel-2, PlanetScope images, KH-9 Hexagon stereo pairs, and SRTM DEM in Hunza Valley in the Central Karakoram [168]. The glacier surging resulted in river damming and a small GLOF damaging a part of the Karakoram Highway. In 2016, 2017, and 2019, three individual ice-rock avalanches in the Petra Pervogo range in the Pamir mountains, were due to glacier detachments that traveled down to the Rasht Valley and studied using very-high-resolution images such as Planet, a worldview in addition to Keyhole, Sentinel-2, Landsat, and ASTER data [7,169]. In 2021, the Chamoli rock–ice avalanche hazard in India was observed by very-high-resolution images of 0.3–0.7 m (WorldView-1,2,3; EO-1, GaoFen-1, DigitalGlobe/Maxar GeoEye-1, Planet SkySat-C), and similarly via high-resolution satellite images by 1.5–2.5 m (Airbus SPOT-7, Cartosat-1) between 2015 and 2021, to generate multiple temporal DEMs for calculating the volume of detached rock and ice [5]. The massive rock and ice avalanche killed more than 200 people as well as destroying four hydropower plants. The Ronti glacier collapse was preceded by thickening processes observed using DEM differences from 2005 to 2020 [63]. In addition to these described icefalls at steep hanging glacier fronts, surge-like glacier detachments appear more frequent than usually reported, along with the potential of low-angle glacier soft beds failing catastrophically [7]. Permafrost degradation and rock glacier creeping increase the likelihood of ice detachments, landslides, avalanches, or rock falls from unstable slopes [11].

Ice/snow avalanches, together with catastrophic snow/ice/rock flows, have been mapped manually or semi-automatically using high and very-high-resolution optical images [5,7,170]. Supervised classification has been used in snow avalanche mapping based on Landsat-8 images with in situ data [171]. Machine learning has been applied in automatic mapping of snow avalanches from Sentinel-1 images in the Western Tian Shan [172]. Additionally, manual surface features tracking was combined with automatic mapping approaches of glacier surges for avalanche identification according to combination usage of optical and radar images and DEMs [169].

Ice/rock avalanches, debris flow, glacier calving, and glacier melting contribution may be the primary triggers of HMA GLOFs, which have presented with increasing frequency since 1980 [173].

Glacier lakes in HMA were investigated using multiple spaceborne optical images [174]. For example, combined usage of manual visual interpretation and normalized difference water index (NDWI) maps were applied to set up a glacier lake inventory using Landsat images in 1990 and 2018 [175]. An NDWI-based model was also used to derive a near global, multi-decadal glacier lakes database using Landsat images of ETM+, MSS, OLI, and TM on the Google Earth Engine (GEE) platform between 1990 and 2018 [10]. A third example details the manual extraction of the 2012 ETM+ images based on the GEE platform, modified NDWI and subsequent manual refinement applied to derive glacier lakes from Landsat 5 TM/7 ETM+/8 OLI from 2008 to 2017 [176]. Most recently, manual interpretation was used in delineating all glacier-feed lakes larger than 0.02 km$^2$ based on Sentinel-2A/B images obtained in 2018, 2020, and 2022 [173].

Primary spaceborne techniques and methods for studying glacier-related hazards are summarized in Supplementary Tables S1 and S4.

## 3. Major Achievements in HMA

### 3.1. Glacier Inventories

Several HMA glacier inventories are available [25,27,28,177,178], with the spatiotemporal glacier coverage and methods presented in Table 2.

The first Chinese Glacier Inventory (CGI-1) [3,179] was generated in the early 2000s based on more than 2000 sheets of topographic map surveys acquired between 1956 and 1984 at a scale of 1:50,000 to 1:100,000, with more than 200 scenes of Landsat MSS/TM images from the 1980s to the 1990s. The second Chinese Glacier Inventory (CGI-2) was produced in the 2010s using optical images from Landsat data, 92% of which were acquired between 2006 and 2010 [28]. Using Landsat series images from 1999 to 2003, over HMA, the Glacier Inventory of GAMDAM (GGI-15) was created in 2015 [27]. Following the GGI-15, the GAMDAM glacier inventory 2018, (GGI-18) was created in 2018 by temporal coverage data from 1990 to 2010 using manual digitization [25]. Following the CGI-2, a 40-year time-span of glacier coverage datasets was generated over the China-Tibetan Plateau in three epochs, namely the Tibetan Plateau Glacier coverage data (TPG), involving TPG1976, TPG2001 and TPG2013, using 263 images from Landsat MSS in the mid-1970s, 150 images from Landsat 7 ETM+ in 1999–2002, and 148 images from Landsat 8 OLI in 2013–2014 [24].

The most frequently used inventory is a single snapshot of global glacier coverage from the Randolph Glacier Inventory (RGI), first released on 22 February 2012 [180], whose contributors involve the glacier database from the Global Land Ice Measurements from Space (GLIMS) [20,181], data collection compilation and production from the GlobGlacier project and Glaciers_cci [182–184], communities from the Cryolist (http://cryolist.org accessed on 25 February 2024), and existing glacier outlines from the world over [183]. The most recent RGI 7.0 released on 19 September 2023, integrates multiple global glacier inventories based on optical images from Terra ASTER, Landsat-7 ETM+, SPOT-5 HRS, Landsat-5 TM from 1 January 1950 to 31 December 2021 [178], which updated the broadly used RGI 6.0 version over the last several years [177] by a working group of the International Association of Cryospheric Sciences (IACS). Glacier inventories require frequent updating due to rapid glacier change and interpretation uncertainties in steep, high-mountain terrain, e.g., mountain shadows, seasonal snow, cloud cover, and insufficient high-quality satellite images from earlier versions [63].

**Table 2.** Major glacier coverage/inventories in HMA.

| Datasets | Temporal Coverage | Region | Number of Glaciers | Glacier Area (km²) | Method | Source |
|---|---|---|---|---|---|---|
| CGI1 | 1979–2002 | China | 46,377 | 59,425 | Manual | [179] |
| CGI2 | 2006–2010 | China | 42,370 | 43,087 | Semi-automatic | [28] |
| TPG1976 * | 1970s | TP China | – | 44,366 | Manual | [24] |
| TPG2001 * | 2001 | TP China | – | 42,210 | Manual | [24] |
| TPG2013 * | 2013 | TP China | – | 41,137 | Manual | [24] |
| GGI15 | 1999–2003 | HMA | 87,084 | 91,263 | Manual | [27] |
| GGI18 | 1990–2010 | HMA | 134,770 | 100,693 | Manual | [25] |
| RGI6.0 | 1998–2009 (major in 1998–2002) | Global | 95,536 | 97,606 | Semi-automatic | [177] |
| RGI6.0 debris-covered ice | 1986–2016 | HMA | – | 75,557 | Semi-automatic | [43] |
| RGI 7.0 | 1950–2021 | HMA | 131,762 | 99,468 | Semi-automatic | [178] |
| KPG 2018 | 2000–2009 | Karakoram and Pamir | 27,800 | 35,520 | Semi-automatic | [185] |
| ICIMOD Glacier Inventory | 2002–2008 | HKH | 54,000 | 60,000 | Semi-automatic | [186] |
| SETPGI | 2011–2014 | Southeastern TP | 7182 | 6440 | Semi-automatic | [187] |

* TPG means the Tibetan Plateau Glacier coverage datasets within China, involving TPG1976, TPG2001, and TPG2013.

Glacier area change on the TP has been studied from space. For example, Hugonnet et al., found that Earth's total glacier area decreased by nearly 10% from 2000 to 2019 [73]. Ye et al., reported a total glacier area decrease of 6.98% on the China-Tibetan Plateau between 1976 and 2013 [24].

### 3.2. Glacier Mass Change

Glacier Mass Balance (MB) change presents significant spatiotemporal heterogeneity across the HMA (Figures 5 and 6). Dramatic glacier mass loss occurred on the Southeast TP, with decreasing trends toward the inland TP and the north and west TP. Hundreds of studies and related datasets have been published, with more than a hundred results summarized in Supplementary Table S9.

In general, the DEM-based geodetic MB in HMA was $-0.18 \pm 0.04$ m w.e.a$^{-1}$ from 2000 to 2016 [72], or $-0.22 \pm 0.05$ m w.e.a$^{-1}$ from 2000 to 2019 [73]. A doubled glacier MB change across the Himalayas was reported from $-0.22 \pm 0.13$ m w.e.a$^{-1}$ in 1975–2000 to $-0.43 \pm 0.14$ m w.e.a$^{-1}$ in 2000–2016 [188]. However, the geodetic MB presented similar results by $-0.21 \pm 0.11$ m w.e.a$^{-1}$ from 1974 to 2000 and $-0.20 \pm 0.03$ m w.e.a$^{-1}$ from 1974 to 2021 based on the same reference DEM in 1974 on the northern slope of the Central Himalayas [70]. Glacier change studies with different reference DEMs may lead to substantial variability in results, for which research on common reference DEMs is encouraged.

However, glacier MB change is significantly inconsistent across different techniques, even when applied in the same geographical region and overlapping time frames. For instance, in Eastern Kunlun (Figure 6a), the MB was $-0.07 \pm 0.07$ m w.e.a$^{-1}$ in 2000–2018 by DEM differencing [59], and $-0.10 \pm 0.05$ m w.e.a$^{-1}$ in 2000–2021 based on the elevation difference between ICESat-2 and NASA DEM [64], whereas it was $-0.49 \pm 0.11$ m w.e.a$^{-1}$ for the period of 2010–2019 using radar altimetry by CryoSat-2 [189]. The second instance, glacier MB in the east Himalayas (Figure 6c) was reported as $-0.33 \pm 0.20$ m w.e.a$^{-1}$ in 2000–2016 [72], and $-0.55 \pm 0.17$ m w.e.a$^{-1}$ in 2000–2018 [59] according to DEM-based techniques, whereas it was $-0.76 \pm 0.20$ m w.e.a$^{-1}$ in 2003–2009 using ICESat footprints [15]. In the third instance, the estimation of glacier MB was $-0.49 \pm 0.19$ m w.e.a$^{-1}$ in Tien Shan in 2000–2008, retrieved with ICESat, and the ice density of $850 \pm 60$ kg m$^{-3}$ [190], which was substantially more negative than $-0.29 \pm 0.07$ m w.e.a$^{-1}$ in 2000–2018 by DEMs [59]. In

the fourth instance, for the same ten glaciers in the Khumbu region by different techniques, the average MB was $-0.45 \pm 0.60$ m w.e.a$^{-1}$ using DEMs calibrated by GPS survey in 2000–2008 [191], $-0.48 \pm 0.05$ m w.e.a$^{-1}$ by DInSAR techniques in 2000–2012 [61], and more negative of $-0.80 \pm 0.52$ m w.e.a$^{-1}$ using multiple DEMs derived from stereo imagery in 2002–2007 [192].

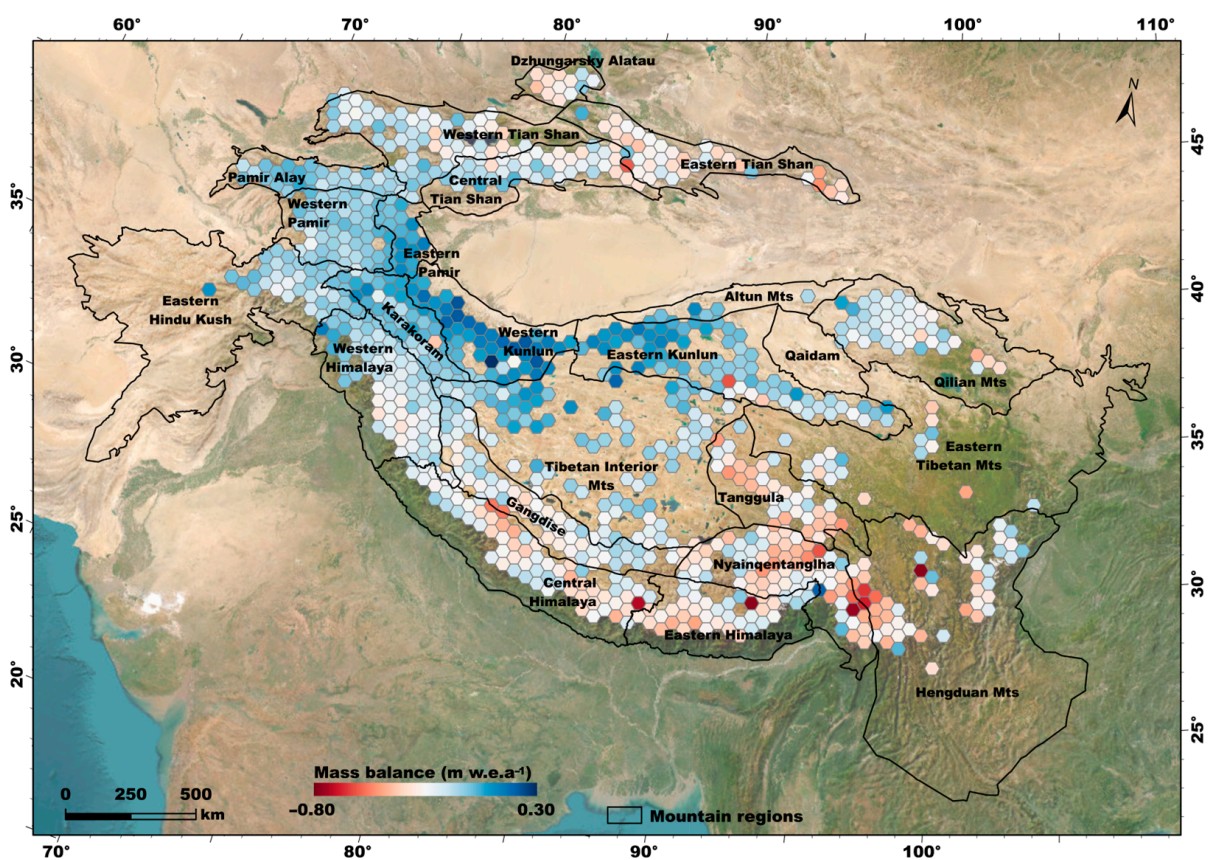

**Figure 5.** Geodetic glacier mass balance (MB in m w.e.a$^{-1}$) between 2000 and 2020 in HMA with the averaged surface elevation differences by 5 km-sized hexagons from the datasets by Hugonnet et al., 2021 [73].

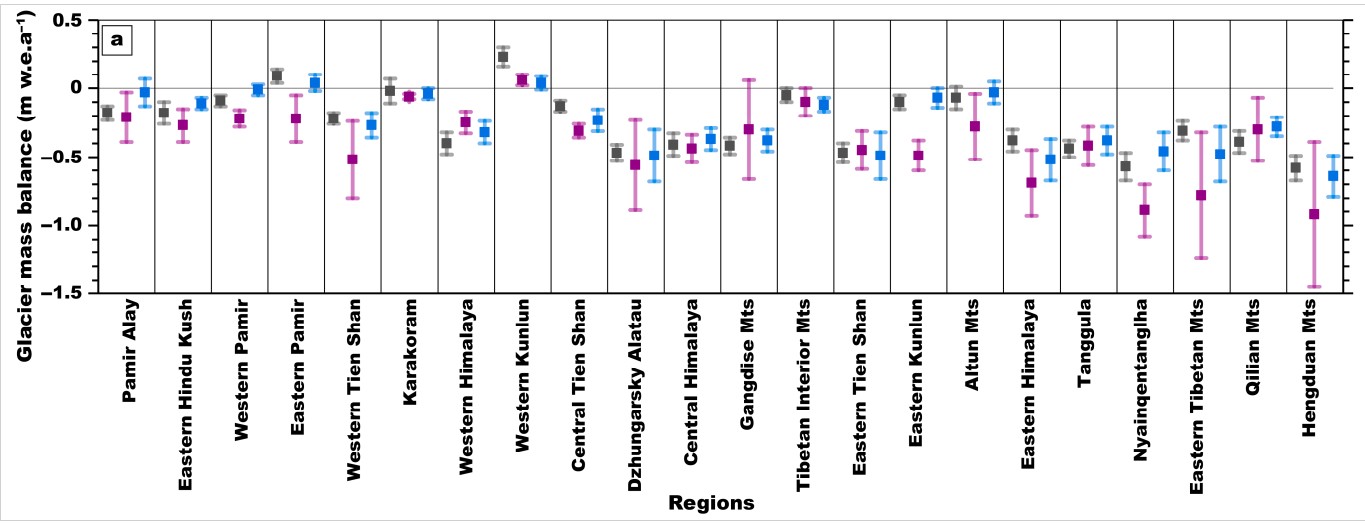

**Figure 6.** *Cont.*

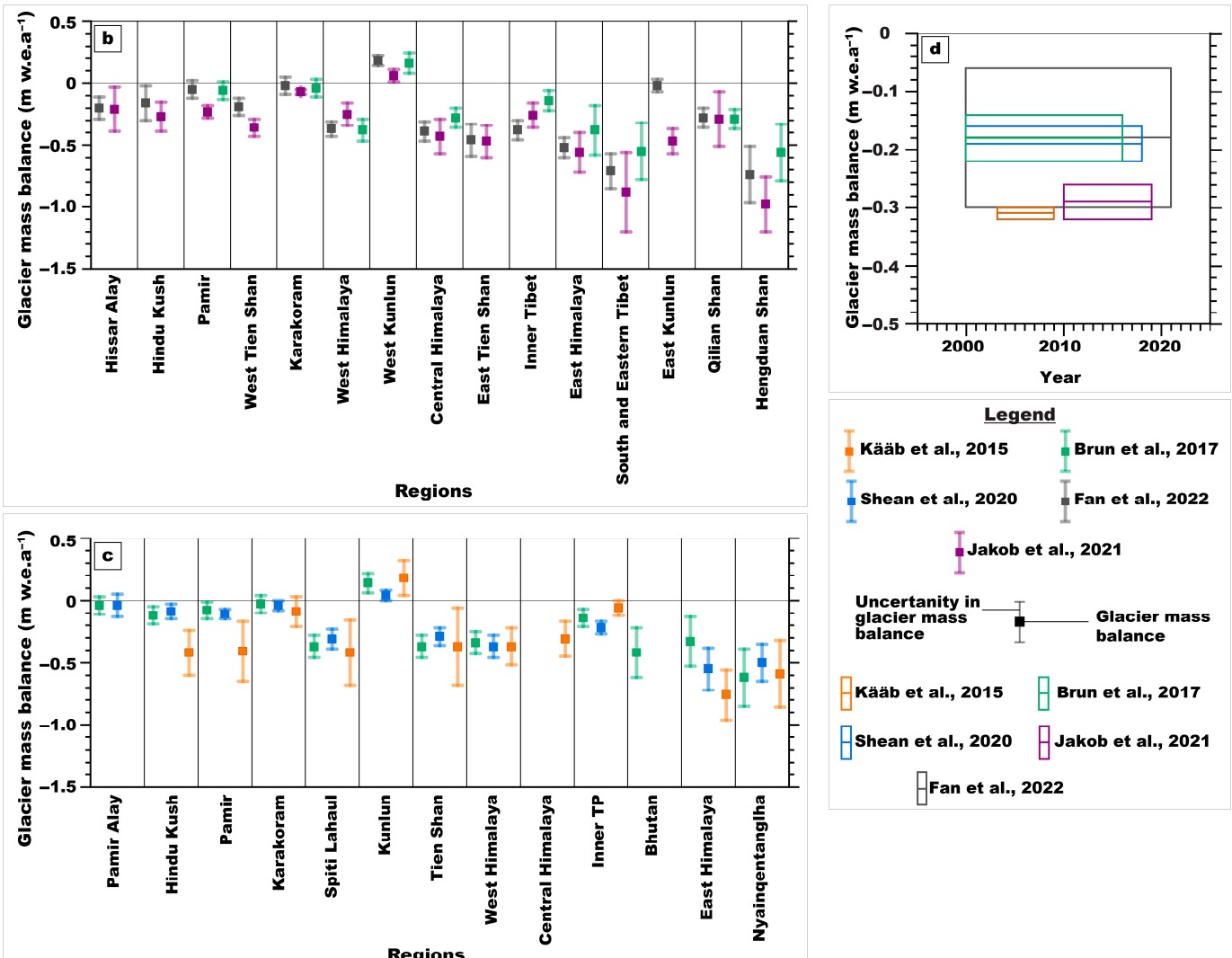

**Figure 6.** Region—wide comparison of glacier-specific mass balance (MB, by m w.e.a$^{-1}$) from five publications aggregated over three different regional boundaries in HMA (MB is marked by spots, its uncertainties are shown by the length of the bars, and different colors represents the data from the corresponding literature). (**a**) HiMAP regions [193]. (**b**) RGI regions [180]. (**c**) Regions by Kääb et al. (2015) [15]. (**d**) The width of the colored bars represents the periods from the five studies across HMA [15,59,64,72,189].

The literature demonstrates that glacier mass changes over HMA are highly variable, with large uncertainties, even with similar techniques (Table 3). For example, based on GRACE data over HMA, glacier mass changed by $-47 \pm 12$ Gt a$^{-1}$ [194] or $-4 \pm 20$ Gt a$^{-1}$ [195]. In comparison, by ICESat/ICESat-2 data during between 2000 and 2021 in HMA, glacier mass change ranged from $-29 \pm 13$ Gt a$^{-1}$ [190] to $-15.6 \pm 10.1$ Gt a$^{-1}$ [196]. Additionally, glacier mass changed from $-16.3 \pm 3.5$ Gt a$^{-1}$ [72] to $-21.1 \pm 1.7$ Gt a$^{-1}$ [73], based on differencing DEMs generated from optical stereo pairs. These differences underscore the complexity of glacier dynamics, highlighting the challenges in establishing consistent studies for glacier change.

**Table 3.** Major glacier mass change reports in HMA in different studies.

| Study Area | Data Used | Time | Glacier Mass Change (Gt a$^{-1}$) | Glacier Mass Balance (w.e.a$^{-1}$) | Reference |
|---|---|---|---|---|---|
| HMA | GRACE | 2000–2010 | $-4 \pm 20$ | $-0.04 \pm 0.20$ | [195] |
| HMA | GRACE | January 2003–September 2012 | $-35.0 \pm 5.8$ | $-0.29 \pm 0.05$ | [197] |
| HMA | GRACE and GRACE Follow-On | April 2002–September 2019 | $-28.8 \pm 11$ | $-0.30 \pm 0.11$ | [198] |
| HMA | ICESat-1, 2 and GRACE/GRACE Follow-On | 2003–2019 | $-28 \pm 6$ | $-0.29 \pm 0.06$ | [199] |
| HMA | ICESat | 2003–2009 | $-29 \pm 13$ | $-0.25 \pm 0.11$ | [190] |
| TP | ICESat | 2003–2009 | $-15.6 \pm 10.1$ | $-0.77 \pm 0.35$ | [196] |
| Pamir–Karakoram–Himalaya | ICESat | 2003–2008 | $-24 \pm 2$ | $-0.31 \pm 0.09$ | [15] |
| HMA | ICESat-2 and NASA DEM | 2000–2021 | $-17.53 \pm 11.36$ | $-0.18 \pm 0.12$ | [64] |
| HMA | Cryosat-2 | 2010–2019 | $-28.0 \pm 3.0$ | $-0.29 \pm 0.03$ | [189] |
| HMA | WorldView-1/2/3, GeoEye-1, and ASTER | 2000–2018 | $-19.0 \pm 2.5$ | $-0.19 \pm 0.03$ | [59] |
| HMA | ASTER | 2000–2020 | $-21.10 \pm 1.70$ | $-0.22 \pm 0.05$ | [73] |
| HMA | ASTER | 2000–2016 | $-16.3 \pm 3.5$ | $-0.18 \pm 0.04$ | [72] |

The different boundaries play an important role in the significant differences in the results of mass change in HMA. For example, glacier MB was obtained by $-0.21 \pm 0.05$ m w.e.a$^{-1}$ based on ICESat and SRTM from 2003 to 2008 in the Hindu Kush-Karakoram-Himalaya (HKKH) [62]. In sharp contrast, a much less negative MB by $-0.14 \pm 0.08$ m w.e.a$^{-1}$ was extrapolated according to the DEM differences between the SRTM DEM and SPOT5 DEM in 9 study sites of the Pamir-Karakoram-Himalaya (PKH over $1999-2011$ [14]. Compared to the HKKH, the PKH region has been significantly extended toward the east (Hengduan Shan, China) and the west (Pamir, Tajikistan) [14]. Another illustrative example is in the East Himalayas in 2000–2016, glacier MB was estimated by $-0.33 \pm 0.20$ m w.e.a$^{-1}$ utilizing the boundary delineated by Kääb et al. (2015) [15], while it was $-0.38 \pm 0.20$ m w.e.a$^{-1}$ within the RGI regions [72]. Therefore, it shows the importance of a standardized definition of study boundaries in cryosphere studies in HMA. Figure 6 illustrates a region−wide comparison of glacier MB studies from the literature over three different regional boundaries.

Glacier changes show an apparent spatiotemporal difference in HMA. Firstly, the most dramatic negative glacier MB was located at Mt. Hengduan in the southeastern TP of $-0.62 \pm 0.10$ m w.e.a$^{-1}$ in 2000–2021, followed by $-0.47 \pm 0.11$ m w.e.a$^{-1}$ in the southern and eastern TP [64]. Secondly, moderate MB occurred at the inland TP and was less negative at the northern HMA, ranging from $-0.47$ to $-0.03$ m w.e.a$^{-1}$. In contrast, an anomalously positive MB of $0.23 \pm 0.13$ m w.e.a$^{-1}$ was detected in West Mt. Kunlun [64]. Moreover, nearly balanced glacier budgets in Pamir and a slight glacier mass gain existed in the Karakoram, with glaciers surging widely distributed [14]. Hence, the Karakoram Anomaly was named for the glacier anomalies in the Karakoram, the Western Kunlun, and Pamir [18].

Glacier thinning rate change with altitude and altitudinal differences in glacier MB were noted in HMA, specifically in the Himalayas. More negative glacier MB occurred at lower altitudes, which had been widely observed [72]. Particularly on the northern Mt. Qomolangma, convincing altitudinal effects on glacier MB were detected [70]. In detail, geodetic glacier MB demonstrated a more negative change with altitude increasing from 5150 m a.s.l. to 5800 m a.s.l., while it became less negative with increasing height above 5800 m a.s.l. from 1974 to 2021. The glacier MB was within $0 \pm 0.1$ m w.e.a$^{-1}$ around 6200 m a.s.l., the equilibrium line altitude (ELA) where the MB reached zero. On the other hand, the debris-covered terminus showed a more negative MB change rate than the exposed glacier ice due to the lower altitude and the debris' higher surface energy absorption [191].

A remarkable difference in glacier MB in HMA was detected according to the aspect orientation. For example, southeastern Tibet's most significant glacier melting areas are

located in the northern flanks, especially the northeast aspect [200]. In sharp contrast, glaciers exhibited weak thinning rates in the southern aspects, particularly those facing the southwest. Additionally, south-facing glacier areas experienced more apparent loss than the north-facing at the Warwan subbasin in the western Himalayan basins [53]. Notably, the windward or leeward aspects orientation of the mountains led to a significant diversity of glacier types at close range [201]. Furthermore, the debris-free glacierized areas shrank faster in the southeastern TP while slower in the northwest TP, in sharp contrast with the rapid expansion of the debris-covered glacierized areas on TP [202].

In addition to the terrain effects, more attention has recently been devoted to glacier change with different termination types. The debris-covered glaciers connected with the glacial lakes presented a more negative MB ($-0.89 \pm 0.36$ m w.e.a$^{-1}$) than land-termination debris-covered glaciers and exposed glaciers ($-0.50 \pm 0.32$ m w.e.a$^{-1}$) in the southeastern TP [200]. Lake-terminating glaciers exhibited a more substantial acceleration in area retreat and mass loss than land-terminating ones in eastern Mt. Tangula between 2000 and 2020 [203]. However, it had a smaller downwasting rate than the land-terminating ice due to terrain effects, sharply contrasting with other Himalayas findings.

*3.3. Snow Cover Extent and Snow Water Equivalent*

HMA snow cover extent products are primarily derived from optical remote sensing images, including MODIS and AVHRR. The National Oceanic and Atmospheric Administration (NOAA) released global snow cover products of daily, 8-day composite, and monthly, with a 500 m resolution from MODIS on Aqua and Terra (Supplementary Table S7). Based on MODIS data, many studies of cloud cover removal were conducted to generate regional daily cloudless products [109], including the snow cover product in HMA from 2002 to 2021 [204]. In addition, a long-term global snow cover extent product (JASMES) was issued by the Japan Aerospace Exploration Agency (JAXA) with a 5 km-resolution of daily, weekly, and half-monthly, created from AVHRR global radiance data from NOAA series satellites (1978–2001) and MODIS (1978–2015) [205]. An entirely gap-free daily snow cover extent in HMA was derived from AVHRR from 1981 to 2019 [206], with quality control, cloud cover detection, snow discrimination, and gap-filling. Thus, a snow cover product was generated by an interactive multi-sensor snow and ice mapping system (IMS) launched by the U.S. National Ice Center (USNIC), integrating various data with satellite images, snow, and ice analysis maps, model data from National Centers for Environmental Prediction (NCEP) and ground observations. Therefore, IMS, combined with snow coverage data and NOAA climate record data, has created a continuous and widely used record of Northern Hemisphere snow coverage since October 1966 [207].

Global remote sensing snow depth/SWE products broadly include the National Snow and Ice Data Center (NSIDC) monthly SWE, AMSR-E/AMSR-2 SWE, and Globsnow SWE (Supplementary Table S7). However, these products were reported to overestimate snow depth in China. Therefore, a locally oriented method was developed to generate a long-term dataset series of daily snow depth in HMA [208]. Based on the dataset, snow depth decreased significantly in HMA from 1988 to 2020 (Figure 7). Globsnow SWE does not cover regions south of 40° N or rough mountainous areas [104]. Therefore, the SWE products' accuracy varies spatially and temporally. When SWE is between 30 mm and 200 mm, the estimated value of GlobSnow is consistent with the ground investigated values. Similarly, NSIDC snow depth products are vulnerable to the influence of microwave "saturation". When SWE is less than 30 mm, the overestimation from GlobSnow is more significant than that from NSIDC products [209]. In contrast, AMSR-E and AMSR2 products agree with ground observations under shallow snow depths (0–10 cm).

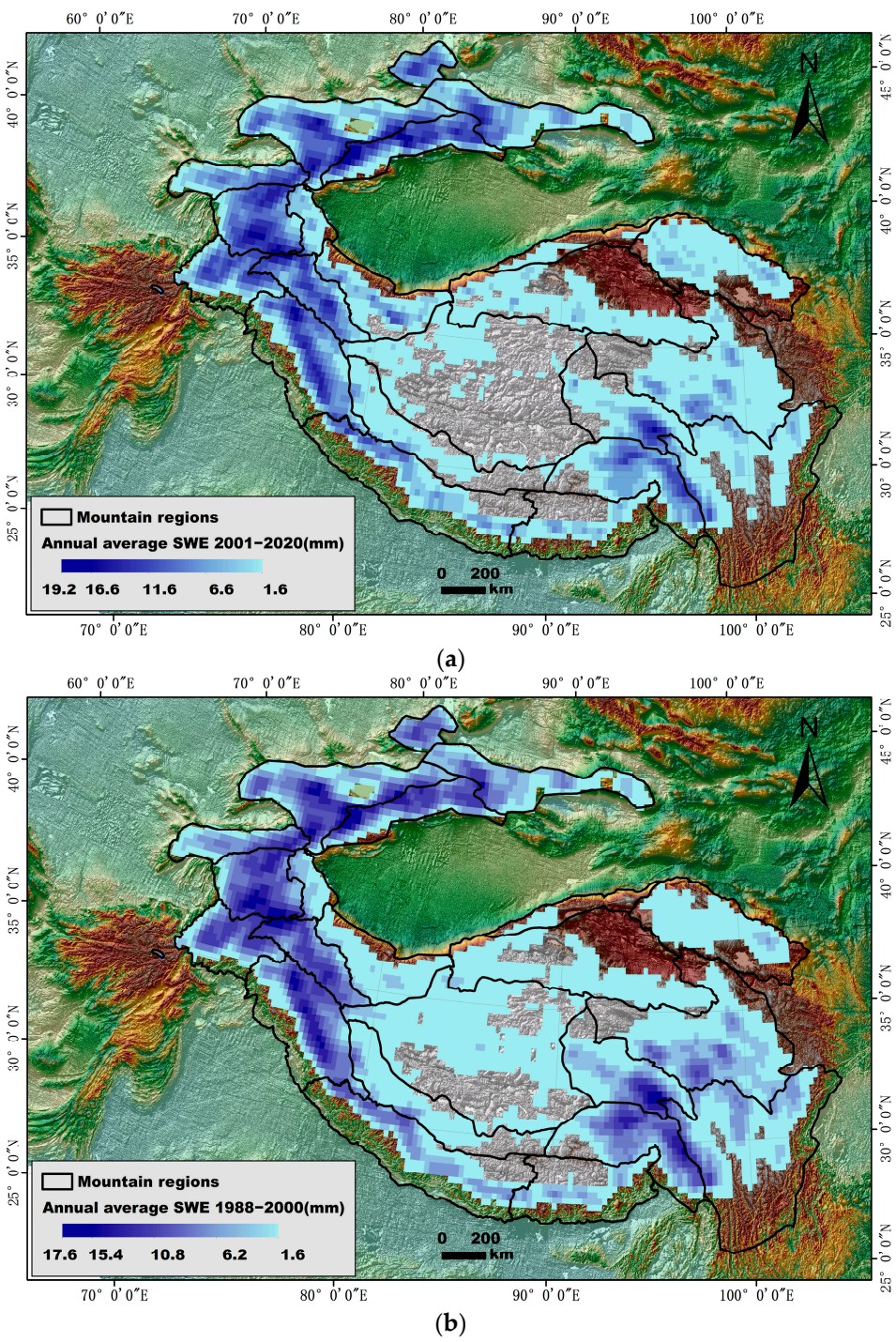

**Figure 7.** Annual average snow water equivalent (SWE) during (**a**) 1988–2000 and (**b**) 2001–2020 (SWE is calculated from snow depth data downloaded from linkage of https://data.tpdc.ac.cn/zh-hans/data/df40346a-0202-4ed2-bb07-b65dfcda9368 accessed on 25 February 2023).

As to the algorithms for snow depth inversion, the support vector regression (SVR) is more effective in retrieving multi-scale snow depth and SWE, combining passive microwave images from space with additional data (i.e., location, terrain, land cover type, snow cover status, and indirect consideration of grain size change) [123]. Although satellite-based passive microwave techniques usually underestimate shallow snow, the strong scattering of deeply frozen soil will commonly overestimate the TP snow depth. Therefore, the inversion accuracy could be improved to 93.9%, with a deviation of 1.03 cm and an RMSE of 7.05 cm, by considering the soil emissivity with spatial heterogeneity to improve the overestimation

of snow cover depth [117]. Specifically, the estimation accuracy is less than 60% when the snow cover depth ranges within 1–3 cm, and the snow cover depth over 25 cm is remarkably underestimated in the Himalayas [117].

### 3.4. Frozen Ground Hydrology

Frozen ground currently accounts for 86% of HMA, including 28% (about $1.27 \times 10^6$ km$^2$) of permafrost and 58% (about $2.59 \times 10^6$ km$^2$) of seasonally frozen ground [127]. The mean annual ground temperature and active layer thickness are the two key climate variables of permafrost that the Global Climate Observing System (GCOS) specified. For reference, Supplementary Table S7 provides the main products of frozen ground using remote sensing techniques.

The soil freeze–thaw cycle is sensitive to ground temperature change. Consistent with other mountainous regions, these physical states exhibit significant spatial variations across HMA. However, direct observations of frozen ground are scarce and unevenly distributed within HMA. The limited boreholes and in situ monitoring on the Qinghai–Tibet Plateau demonstrate that the frozen ground temperature at 10 m depth has been increasing by 0.02–0.78 °C decade$^{-1}$ from 2004 to 2018, with a gradual increase in active layer thickness by 8–39 cm decade$^{-1}$ from 1998 to 2018 [210–212].

Since decadal-long global soil moisture products and freeze–thaw states can be derived from active and passive microwave measurements (Supplementary Tables S6 and S7), these surficial hydrological components are readily available and are among the best-characterized data generated by remote sensing methods in frozen ground regions. Various remote sensing surface and root-zone soil moisture products were evaluated with in situ measurements on TP [213], such as the Advanced Microwave Scanning Radiometer 2 (AMSR2), Soil Moisture and Ocean Salinity (SMOS), and Soil Moisture Active-Passive (SMAP). Several best-performing remote sensing products, such as SMAP Level 4, were identified in terms of Pearson's correlation coefficients and unbiased RMSE [213]. However, limitations were reported in these soil moisture products over the permafrost areas on TP, i.e., all products overestimated the surface moisture, with significant errors in the root-zone soil moisture products during thawing–freezing transitional periods. Conversely, InSAR ground deformation data in 2015–2019, were combined with an independent multivariate statistical method [214]. By isolating the cyclic seasonal component of the surface displacements related to active layer freeze and thaw from other signal components, researchers can infer a net increase in the active layer water content of approximately 8 cm equivalent water thickness and widespread intensification of the subsurface water cycle in the central TP.

Thermokarst lakes, caused by ground subsidence with permafrost thawing, are scattered throughout frozen ground regions and on many debris-covered glaciers in HMA. Based on Sentinel-2 images, Wei et al. identified about $1.6 \times 10^5$ thermokarst lakes across TP, with an area of approximately $2.8 \times 10^3$ km$^3$ [215]. More than 94% of the thermokarst lakes are located in continuous permafrost zones, which is not surprising as permafrost underlies 90–100% of the area. Based on aerial and satellite images, the thermokarst lakes increased in number and surface area for four regions along the Qinghai–Tibet Highway from 1969 to 2019 [216]. That the total number of thermokarst lakes increased from 633 in 1969 to 1580 in 2019, and that there has been significant growth and expansion of lake size is consistent with and attributed to climatic warming and increased precipitation in the region. According to a preliminary assessment from borehole measurements and surficial geology, permafrost on TP stores about $12.7 \times 10^3$ Gt of water equivalent in ground ice [217]. Rock glaciers also contain significant ground ice. For example, the rock glaciers in the Himalayas are estimated to be approximately 52 Gt of ground ice [132]. However, such estimates for HMA are impossible because of the lack of comprehensive rock glacier inventories over the whole of HMA. Rock glacier inventories are presented in Supplementary Table S6, involving Northern Tian Shan/Tien Shan, Western Nyainqêntanglha Range, the Karakorum, Daxue Shan at south-eastern TP, the Poiqu catchment, and the Himalayas.

The melting of ground ice is likely one of the most significant contributors to the hydrological system in HMA. However, estimating and measuring ground ice content and its temporal changes remains challenging via remote sensing, as previously discussed. Despite a few studies making a quantitative effort to link ground subsidence to ground ice loss, estimations on ground ice loss have been explored recently using InSAR-derived subsidence (Supplementary Tables S1, S6 and S7). For instance, the widespread ground subsidence of up to 2 cm/year is revealed due to permafrost thawing based on InSAR, including Envisat and Sentinel-1 data spanning 16 years over a large permafrost area in the northern TP [218]. The ground ice loss rate is estimated to be approximately $57 \times 10^6$ m$^3$ year$^{-1}$ in the permafrost degradation region of the Selin Co basin from 2017 to 2020 by equating the multi-year trends in surface subsidence with the volume change in ground ice, which contributed approximately 12% of an observed increase in lake volume [219].

### 3.5. Lake Ice Change

Lake ice studies have been emphasized in the cryosphere over recent decades [150,220–222], with many large-scale lake ice products derived from spaceborne data covering the HMA (Supplementary Table S10), the North Temperate Zone, and even the Northern Hemisphere [223,224]. According to ground observations of 39 lakes in the Northern Hemisphere over the past 150 years, the lake ice phenology changed significantly with delayed freezing dates and earlier melting dates [222], consistent with recent studies [223,225]. However, in situ lake ice records are so rare and costly that observations from space are now broadly applied for lake ice monitoring [152], including the remote sensing of optical images, passive microwave (radiometers), and active microwave (SAR).

Optical satellite-based techniques are widely utilized to detect lake ice phenology in mid-latitude areas. For instance, based on stratified random samples from Sentinel-2 and PlanetScope, seasonally frozen phenology was found in 41% of global inland lakes, with lake ice cover decreasing significantly from January to March, with the most pronounced decline in Central Asia and East Asia [226]. The advantages of daily revisits by MODIS data facilitate the observation of lake ice change phenology on TP with high temporal resolution [155,221,227]. Some alpine lakes were reported unfrozen in winter, changing from perennial to seasonal lake ice. In addition, Guo et al. evaluated the uncertainties of three optical remote sensing methods and discussed the relationship between climatic factors and lake ice phenology in TP [228].

Passive microwave successfully generates regional and global lake ice monitoring without being influenced by clouds and nighttime. For example, Du et al. investigated the lake ice phenology using the daily brightness temperature data from AMSR-E/2 from 2002 to 2015 in the Northern Hemisphere [224], which presented shorter lake ice duration for 60% studied lakes, especially those lakes at higher latitude. In addition, Cai et al. achieved a lake ice phenology dataset using SMMR, SSM/I, and SSMIS data for 56 large lakes from 1979 to 2020 in the Northern Hemisphere [223]. Moreover, a lake ice dataset was retrieved from the passive microwave technique, predicting future lake ice phenology in different Representative Concentration Pathway (RCP) emission patterns in TP [229].

Lake ice thickness can also be acquired from active microwave techniques because of the different backscatter coefficients between water and ice [152]. For example, a long-term dataset of lake ice thickness was generated using satellite pulse-limited radar altimeters (TOPEX/Poseidon and Jason-1/2/3) for 16 large lakes/reservoirs over the past three decades in the Northern Hemisphere [158].

Additionally, it is efficient to study lake ice by comprehensively utilizing satellite-based techniques and models. For example, lake ice thickness was simulated for 1313 lakes in the Northern Hemisphere between 2003 and 2018, driven by ERA5 data (daily air temperature, humidity, wind speed, and precipitation) and MODIS (four MODIS LST products, one MODIS albedo product), disclosing the annual maximum ice thickness was

0.63 ± 0.02 m [158]. Looking forward, changes in lake ice thickness are projected for 2071–2099 under different RCPs.

### 3.6. Database of Glacier-Related Hazards and Inventories of Glacier Lakes

The first snow and ice avalanche database in HMA, HiAVAL, captured 681 events collected from available records or previous studies between 1972 and 2022 [13]. In addition, eight actual large-volume, low-angle glacier detachments in HMA were compiled and investigated [7].

The first comprehensive GLOFs inventory in HMA documented that 697 individual floods occurred between 1833 and 2022, as compiled from peer-reviewed papers, news articles, reports, books, and communications [12]. To date, several glacier lake inventories in HMA have been derived from optical images. For example, a nearly global multi-decadal glacier lake inventory has been derived between 1990 and 2018 [10]. Other glacier lake inventories in 1990 and 2018 were integrated across HMA [175]. The third example of glacier-lake inventory in HMA (Hi-MAG) was created from 2008 to 2017 [176]. Most recently, all glacier-fed lakes larger than 0.02 km$^2$ were compiled into inventories in 2018, 2020, and 2022 [173]. The database linkage is presented in Table S11.

## 4. Remote Sensing Applications in Cryo-Hydrological Modeling

There are significant research challenges for ground-based observations and the hydrological processes in the cryosphere in HMA related to the steep high mountain terrain and logistical difficulties. As such, in situ data scarcity hampers the application of physics-based models for simulating cryospheric processes. Alternatively, remote-sensing-based models offer the potential for the physical understanding and quantitative assessment of HMA-scale cryosphere changes. Therefore, combining numerical models with various remote sensing products and climate data is indispensable for cryosphere modeling in HMA. Consequently, numerous efforts have been undertaken to develop cryospheric models with different complexities over the past decade, deepening the understanding of current cryospheric changes and projected trends. Remarkably, many satellite-derived products are freely available, which is crucial for model forcing, calibration, and validation in cryospheric research.

### 4.1. Glacier Melt Modeling

#### 4.1.1. Models

Glacier meltwater is a special contributor to river runoff in HMA with seasonal variations well defined. Study approaches include temperature-indexed models, which are widely used to simulate glacier melt as they require only temperature as an input variable, i.e., air temperature, assuming unchanged glacier geometry within the short term. However, models suffer from significant uncertainties due to oversimplifying key dynamics, inaccurate boundaries, initial conditions, and other energy balance terms and their temporal evolution. Hence, it is essential to disclose the mass redistribution process caused by glacier change to project the multi-decadal glacier evolution [230]. Studies now use glacier models coupled with MB and ice flow dynamics for long-term projections of glacier evolution.

The Global Glacier Evolution Model (GloGEM) [231] is commonly adopted to estimate glacier meltwater [232]. Within the Glacier Model Intercomparison Project (Glacier-MIP), glacier mass change projections in HMA have been presented using six glacier evolution models (including the GloGEM) [233]. Further, the Open Global Glacier Model (OGGM) [234] is a global-scale open-source model for simulating glacier behavior and evolution, combining physical principles with observational data to calculate glacier MB, ice dynamics, and glacier-fed runoff [234]. The input data used in OGGM include glacier outlines, topography, and climate validation data. The Python Glacier Evolution Model (PyGEM) is an open-source model designed to simulate glacier MB and volume changes,

with a minimum dataset of inputs including a glacier inventory with attributes of ice area, ice thickness, and climatic variables—specifically, temperature and precipitation [235,236].

### 4.1.2. Input Parameters

In recent years, three available global-scale products related to the debris-covered extent and thickness have provided essential inputs for modeling [43,202,237].

Accurate information about glacier geometry (i.e., glacier outline, surface terrains, and bed topography) is essential to improve the energy balance of glacier models.

Firstly, glacier outlines (clean and debris-covered) are generally taken from regional or global glacier inventories [24,25,27,28,177,180]. The RGI is broadly utilized as a reference database for estimating glacier-specific MB and for projecting future glacier changes. However, significant uncertainties are noted in the RGI when delineating glacier divides and glacier tongues and determining glacier terminus types.

Approximately 12.5% of glacier areas in HMA are covered with debris [43], while it occupied 16.9% of South Asia East [202]. Therefore, the accurate extent and thickness of debris on glaciers will improve the estimation accuracy of glacier mass loss [235].

It is essential to determine the glacier terminus types (e.g., lake terminating or land terminating) in HMA by integrating the inventories of glaciers and glacial lakes [175,238,239]. Since the 1990s, the world's glacial lakes have rapidly grown by approximately 48% due to climate warming and glacier shrinkage [10]. Furthermore, the growth of glacial lakes has facilitated much more glacier mass loss and increased flow speeds due to glacier calving and basal sliding in HMA [239–241]. Therefore, a comprehensive glacier inventory, with accurate information about the ice divide, proglacial lakes, and debris cover, is crucial to accurately estimate the glacier melt contribution to downstream rivers.

Surface topography is obtained from satellite-derived elevation products with relatively low uncertainty. Currently, several freely available global DEMs exist, e.g., the SRTM DEM, TanDEM-X 90 m DEM, ALOS World 3D DEM, ASTER GDEM, and the Copernicus DEM (Supplementary Table S3). However, it is challenging to determine the bed topography because of the lack of ice thickness measurements. Nevertheless, direct glacier thickness measurements are costly and time-consuming [242], including ice core drilling into the glacier bottom, seismic reflection, gravity, radar echo sounding, or ground penetrating radar measurements [243]. Hence, researchers have proposed various methodologies to model ice thickness based on satellite-derived surface characteristics (i.e., surface elevation, slope, outline, and horizontal surface velocities) [232,244]. Additionally, the model-based global glacier thickness could be retrieved from glacier velocity products [19]. Over time, researchers developed different ice thickness estimation models to infer the spatial distribution of glacier thickness based on surface parameters (elevation and slope) and glacier flow behaviors [232,244]. However, the accuracy of model-based glacier thickness depends heavily on the slope and ice surface flow velocity. Further, a glacier surging phase may disturb ice thickness estimation [19].

Several global-scale glacier thickness products are available [19,232,245]. However, significant uncertainties remain in various thickness products from different inversion approaches and data inputs. For example, the ice volume in HMA is estimated to be 37% higher by Millan et al. [19] than that from Farinotti et al. [232]. Therefore, the uncertainties of inferred ice thicknesses should be considered in future glacier evolution modeling, which will affect the assessment of the peak water timing and water pressure on residents.

### 4.1.3. Calibration and Validation

At the basin scale, the parameters of glacier melt models are generally calibrated by runoff datasets [201]. This calibration procedure, however, may produce similar results using different parameter sets, which is named the equifinality problem. Therefore, a model should be calibrated against multiple response variables to reduce unrealistic simulation results. For example, the energy balance glacier model could be calibrated using satellite-derived snowline altitude and geodetic glacier-wide MB.

It is vital to evaluate model performance to lower the uncertainty of future glacier change modeling. Model validation should be independent of the calibration procedure [230]. For instance, the runoff dataset should not be used for validation if it has been utilized for calibration.

### 4.2. Snowmelt Modeling

Snowmelt in HMA is generally simulated by a temperature-indexed model similar to glacier meltwater modeling [246,247], the Snowmelt Runoff Model (SRM) [248], and the Cold Regions Hydrological Model (CRHM) [249]. However, the scarcity of local data prevents the usage of physically based multi-layer models at the scale of HMA, despite their higher accuracy at local scales when more data are available. Instead, satellite-derived snow products (e.g., snow cover extent and snow depth) are usually applied to calibrate and validate snowmelt models. Although passive microwave satellite-derived SWE datasets are available in HMA [250,251], the relatively coarse resolution and lower accuracy in mountainous areas limit their usage in snowmelt modeling [246].

### 4.3. Permafrost Hydrological Modeling

Permafrost regulates the interaction between surface water and groundwater. Permafrost thaw impacts hydrology by increasing the permeability of partially frozen material and releasing the meltwater of subsurface ice [252,253]. However, the impacts of permafrost hydrology on HMA remain unclear, and permafrost hydrology is not well integrated into the current cryo-hydrological models. Additionally, simulated river runoff generally neglects the contribution from permafrost thaw.

Rather than the large HMA scale, most ground-based permafrost observations were carried out on TP [254,255]. Remarkably, there are significant uncertainties regarding the results of permafrost distribution because of the scarcity of field-based observations in mountainous areas of HMA. For example, Schmid et al. provided a first-order estimation of permafrost extent and distribution utilizing the rock glaciers (i.e., one proxy of permafrost existence) in the Hindu-Kush-Himalayan region [256]. The first near-global rock glacier database was compiled using a meta-analysis methodology with likely larger overall uncertainty [134]. However, the permafrost extent is approximately ten times larger than the glacierized area [257]. It is well known that the permafrost extent was overestimated by the approaches using rock glaciers as a proxy due to the ground-cooling effects of rock glaciers [258].

Unknown parameters such as the permafrost distribution and freeze–thaw processes are difficult to retrieve from space in the Himalayas [201], which prevents the basin-scale application of permafrost models and the assessment of the contribution of permafrost to river runoff. These data limitations regarding permafrost hydrological processes will exist for the long term, even with improvements in field measurements and remote sensing data observations [253].

### 4.4. Lake Ice Modeling

Mathematical models, e.g., LIMNOS (Lake Ice Model Numerical Operational Simulator), were created to define lake ice change in historical period or for lake ice predictions in a warming climate [259,260]. These models describe lake ice changes based on lake thermodynamics, including lake energy balance [261–264]. In addition, the simplified process-based models, with the timing of lake ice freezing and breakup [265], may perform as well as more complicated models [266].

## 5. Discussion
### 5.1. Summary of Cryosphere Studies from Space

Satellite-based techniques have innovated cryosphere observations since the 1960s. Over the last 60 years, advances in these remote sensing techniques have achieved significant progress in cryospheric studies.

Multi-spectral images from spaceborne optical sensors are popularly applied in cryosphere observations. In general, the multi-spectral bands involve visible (VIS), near-infrared (NIR), short-wave infrared (SWIR), and thermal infrared (TIR). First of all, bands of VIS/NIR/SWIR are broadly used in differentiating land surface, such as deriving glacier coverage [24,185,267], glacier surface velocities [74,79], glacier surges [268], supraglacial debris cover [269–271], rock glacier identification [135], snow cover extent/fraction/days [108,109,204,205,272], snow and ice albedos [107,273], frozen ground distribution [274], periglacial geomorphology mapping [275], rock glacier areas [132,138,256], glacial lakes [10,173], Aufeis [146], glacier collapses [167], and lake ice phenology (e.g., date of lake ice appearance, break up, disappearance, extent/coverage/area, and existence duration) [221]. Very-high-resolution optical images are also widely applied in observing cryospheric hazards involving ice and rock avalanche [5], postseismic and coseismic landslides, and landslide-impacted lakes [276], glacier detachments [7], and glacial lake outburst floods [12]. Moreover, the TIR bands are usually applied in retrieving LST products for lake ice phenology [221], deriving MAGT for studying the thermal state of permafrost [126], or frozen ground distribution [274], etc. In addition, optical stereo pairs, such as Terra/ASTER, ALOS/PRISM, SPOT1-4/HRV, SPOT5/HRG, ZY-3-01/TAC (Supplementary Table S2), are used to generate DEMs over decadal periods to map terrain topography or derive surface elevation changes for studying geodetic glacier MB [23,58]. In particular, sub-meter resolution optical stereo images (e.g., Worldview/WV) have provided more contrast details without saturation problems in snow-covered areas and higher vertically precise DEMs [23].

Microwave remote sensing techniques are good for viewing the land surface at any time of day, regardless of cloud cover. Among them, passive microwave brightness temperature data from space have been widely applied in snow cover [277], snow depth [112,278], SWE [209,279], ground surface freeze–thaw state [124], permafrost map [129], permafrost active layer thickening rate [280], lake ice phenology [159], lake ice thickness [281]. In contrast, active spaceborne microwave techniques provide higher spatial resolution and facilitate the study of subsurface structural properties by penetrating signals through the land surface, e.g., ice, snow, and frozen ground. The most frequently used active microwave data are SARs, which have been applied in mapping glaciers [185,282], retrieving debris-covered glaciers [46,51], mapping rock glaciers [283], snow cover [284], lake ice coverage and thickness [152,285]. In addition, InSAR has been widely applied in frozen ground deformation [214,218,286], as well as for an inventory of rock glaciers [133] and rock glacier creep speeds [139]. Moreover, bistatic SAR interferometry has been broadly used in DEM generation [61], glacier surface velocity [80,82], snow depth/SWE [287], and the active layer thickness of permafrost [280]. In addition, DInSAR techniques have worked very well in studying glacier surface elevation difference and glacier geodetic MB [61], glacier surface velocity [80], rock glacier creep speed, and movement of glacial and periglacial processes [139]. Moreover, satellite altimeters, e.g., SAR or Lidar altimetry, have been very popular in observing glacier MB [64,189,196], lake/river ice thickness [288], and lake/river water level [289,290].

Based on the satellite datasets described above, vast quantities of cryospheric data have been retrieved by various methods (Supplementary Tables S1 and S4–S10). In addition, the traditional visual interpretation and semi-automatic methods, such as band ratio [267], band-based indexes [291], or supervised classifications [282] are frequently carried out to obtain the components with manually selected and optimized threshold values for individual scenes. Object-based image analysis was applied in surface segmentation, e.g., glacier mapping [37,292], definition of debris-covered glaciers [292,293], and rock glacier detection [141]. Machine/deep-learning classifiers continue to develop and find application in identifying cryospheric factors [35], e.g., glacier delineations [34,294–297], debris-covered glacier mapping [45–48,298], snow cover and depth [35], rock glacier distributions [141], etc. However, quantities of visual interpretation and manual work remain indispensable in pre-processing and post-processing, especially for outlining surge-type glaciers [16].

Combined use of spaceborne data from multiple sensors emerged over the last decade. For example, combining SAR and optical images involving ALOS PALSAR and SPOT have been applied in active rock glaciers mapping [283]. Application to glacier surging provided data for glacier surface velocities, DEM differencing, and visual interpretation based on these spaceborne optical and InSAR data [89]. Combining models and satellite-based data offers several advantages, for example, increasing the accuracy of snow depth based on the AMSR2 TB and the SNTHERM simulations [115]. In another example, debris-covered glaciers were delineated from joint usage of Sentinel-1/-2, Landsat 8 (TIR), and DEM by deep-learning algorithms [46,51].

In summary, large amounts of available satellite data have significantly compensated for high-cost field investigations historically applied to study cryosphere changes and related hazards. Remote sensing techniques and data-retrieving methodologies are summarized in Supplementary Tables S1, S2, S4, S6 and S8. Major products from space in HMA are presented in Supplementary Tables S1, S3, S5, S7 and S9–S11.

*5.2. Hydrological Effects of Warming Cryosphere in HMA*

Cryospheric meltwater plays a crucial role in glacierized catchments, especially for upstream tributaries located close to glacier areas in mountainous regions [299], supporting the lives of approximately one billion people in HMA [201]. Therefore, cryosphere changes in catchments are increasingly essential for climate–hydrology studies. For example, a recent study discloses that glacier mass loss accounts for $19 \pm 21\%$ of lake volume increase on the entire TP [300].

The spatiotemporal variation in glacier melt contribution to downward runoff/lakes is remarkable [301]. Specific to location, glacier melt contribution varies across the whole of HMA. For example, the major western basins in HMA (Amu Darya, Syr Darya, and Indus basins) significantly depend on ice and snowmelt in summer (June–August), whereas rainfall dominates summer runoff in the east TP (Ganges and Brahmaputra basins) [302]. Specific to the temporal component, glacier meltwater contribution varies in different periods, even for the same catchment. For instance, glacier contribution to runoff was above 50% from 1974 to 2006 [58], sharply contrasting with noted contribution changes between 30% and 73% in different epochs from 1974 to 2021 [70], based on geodetic glacier MB with the in situ hydrological investigation at the Rongbuk catchment in the Central Himalayas.

Recent studies have attempted to separate meltwater's runoff composition from glaciers, snow, and permafrost. However, apparent differences exist in the assessments of individual contributions. For example, Yang et al. partitioned the streamflow in the Nu River watershed, noting a 3.2% contribution from glaciers, 13.4% from snow, and 0.8% from permafrost [303]. The meltwater contribution to the streamflow from the snow was more than that from glaciers in the large glacierized areas (e.g., Hunza basin in Pakistan Karakoram region [304]). Snowmelt also dominated the western Himalayas' contribution, generally by three to five times larger than glacier meltwater. Similarly, the total meltwater contribution from snow and glaciers to streamflow was approximately 69.1% in Indus, 84.5% in Amu Darya, and 59.6% in Syr Darya [246]. Nevertheless, glacier melting could last for the entire melt season when snowpacks are depleted in the late summer [302]. Thus, glacier meltwater becomes crucial during drought periods when snowmelt and rainfall are absent [305].

Predicting the maximum (peak water) timing of glacier runoff in HMA is complex [301]. Most studies indicate that peak water has been reached or will be reached in the next two to three decades [299]. Other researchers report that the peak water of glacier runoff will occur before 2050 in the Ganges and Brahmaputra river basins, while the peak water of the Indus will occur after 2060 due to the giant ice volume fed by westerlies [235]. Despite the uncertainty in the timing of the peak and its subsequent decline rate, runoff will decrease coincident with glacier shrinkage after glacial runoff peaks in the Himalayas and Karakoram [306].

Estimating the glacier/snow/permafrost contribution to river hydrology is fraught with considerable challenges, especially in projections through the 21st century [201]. These challenges involve the comprehensively unknown baseflow, the significantly insufficient observations for vast area permafrost (e.g., freeze–thaw process, degradation), inaccurate glacier outlines, and broadly diverse estimates of glacier loss (both area and mass), the nonlinearities of surface water in controlling glacier melting, the poorly understood mechanisms of debris-cover effects on glacier melt, the ignorance of debris-cover evolution, the considerable uncertainties of snow depth and SWE.

*5.3. Major Problems and Outlook in Spaceborne Cryospheric Monitoring*

Many challenges remain in spaceborne cryospheric monitoring, including limited techniques and methodologies, knowledge gaps (e.g., ice flux, surface sublimation on snow/glaciers), insufficient high spatiotemporal resolution data over the long term, poor data quality, data incompatibility from different techniques, inadequately known parameters in existing models, over parameterization, lack of ground-based verification, inaccurate and inconsistent estimated results, and substantial uncertainties, and related.

The inaccurate outlines of glaciers, especially the margin definition of debris-covered glaciers, lead to inconsistent estimates of glacier coverage, thickness, and ice mass change with its hydrological effects. Specifically, the different approaches and the vast amount of short-term spaceborne measurements without ground-based verification (e.g., bedrock topography, ice rheology, glacier geometries, ice motion, glacier thickness, and debris-cover thickness on ice) impose substantial uncertainties in studies on catchment-wide or HMA-wide estimation of ice reservoirs, resulting in 34% ice volume difference in the Himalayas [19,232], or 20–30% glacier mass change discrepancies [23]. Consequently, a definitive consensus on glacier contribution has not been achieved, complicating the measurement of streamflow magnitude, peak water timing, and future glacier change projections. Additionally, the high uncertainties in glacier change challenge our understanding of the hazard cascade from the collapse of rock and ice [5] or potential hazards by GLOFs due to glacier lake enlarging by ice melting or glacier surging [173,307].

As glacier coverage is one of the critical inputs for glacio-hydrological models to better define and understand glaciers' hydrological effects, it is vital to update glacier inventories and their coverage using state-of-the-art methodologies including a focus on rapid glacier change, especially providing the margins of debris-covered ice. To compensate for the insufficient ground-based observations, glacier downwasting studies in HMA can be cross-verified by available independent DEMs or surface elevation difference datasets from InSAR, photogrammetry optical stereos, and laser altimetry multiple techniques. Moreover, the numerous separate datasets could embed short-term measurements into a comprehensive long-term study with different techniques and methodologies. Additionally, the uncertainties would be lowered to a large extent with a common-based reference DEM. Finally, geodetic glacier mass balance estimations could be cross-verified by multiple parallel independent datasets in various long periods [70].

Frozen ground is fundamentally challenging for remote sensing techniques because of vast areas' complex subsurface freeze–thaw conditions. In addition, substantial knowledge gaps exist due to the subsurface inabilities of spaceborne sensors and the paucity of in situ observations, posing a critical challenge in understanding permafrost hydrology. The freeze–thaw process with energy transfer and water flux in soil involves complex interactions between ground ice and sub-permafrost, intra-permafrost, and supra-permafrost at regional or continental scales [253]. For MODIS LST products, the relationship between canopy and ground surface temperature is unclear [255]. Therefore, additional advanced satellite technologies and observing methods are critical for long-term continuous measurements. These approaches may reduce the estimation errors in quantifying soil moisture, the magnitude of freeze–thaw ice, active layer thickness, permafrost degradation, baseflow process, the accurate ground surface temperature in densely vegetated areas, underground soil moisture, ground ice change, and permeability of bedrock. Inadequate monitoring and

unknown underground water flow could be supplemented with permafrost hydrology modeling and spaceborne remote sensing data. Thus, it is possible and imperative that we deepen our understanding of permafrost dynamics and how they are impacted by climate change, allowing us to predict better and mitigate their effects on the environment and infrastructure in HMA.

Specific to spaceborne snow studies, scattering is a critical problem for both passive and active remote sensing methods. Passive microwave satellite-based remote sensing is the most effective technique to monitor long-term snow cover depth at large scales. However, substantial errors (above 50–100%) have been detected in estimating snow depth and SWE for deep snow, especially when using algorithms for microwave radiometers under wet snow circumstances with coarse resolution [308]. Active microwave techniques (e.g., SAR) are suitable for estimating snow cover depth at a watershed scale with high spatial resolution but is limited by the penetration depth and frequency range. Specifically, higher-frequency SAR sensors have limited penetration through deep snow cover.

Lower frequency SAR has a greater penetration depth through the snow with lower spatial resolution. Hence, a multi-frequency polarimetric SAR system may help identify the spatiotemporal changes in SWE. The estimation accuracy of mixed pixels is improved by combining passive microwave and optical remote sensing. However, the problem of microwave signal saturation remains due to snow depth and multiple scattering between steep slopes in rough mountainous areas. Therefore, characterizing the scattering process is essential for reliable remote sensing results. Furthermore, remote sensing of snow cover is challenged by the impacts of diversified snow properties (such as density, crystal shape, and snow grain size) on the signals received from satellite sensors. In addition, it is difficult to obtain accurate and consistent snow data from space because of various environmental factors (e.g., temperature, dust particles like black carbon, and wind).

It is essential to analyze data uncertainty and develop advanced techniques to improve the accuracy of snow studies. Generally, the focus should incorporate ground-based observations and modeling to validate remote sensing data and apply a multi-method approach to improve the estimation accuracy of snow depth. The multi-methods include: (1) features summarization of snow cover via snow pit observations [112,278]; (2) snow process models [114]; and (3) combined models of land surface and snow. Overall, new remote sensing techniques are required to reduce data uncertainty to better capture the variations in snow properties.

Spaceborne research on lake ice thickness is inadequate in HMA. The availability of diurnal data with high spatial resolution is one challenge. Optical remote sensing (e.g., MODIS) is also challenged by data gaps due to cloud cover. Further, spaceborne passive microwave (e.g., SMMR, SSMIS, SSM/I, and AMSR-E) is limited by low spatial resolution. On a positive note, active microwave data are available in any weather/day with a comparable spatial resolution to optical images. However, disadvantages of active microwave continue to limit its usage, such as involving less spatial coverage, longer revisit intervals, selected beam modes, more complex preprocessing steps, the complex impact of snow on lake ice, and the difficulties in retrieving lake ice thickness due to the knowledge gaps on the complex surface scatter of microwave signals [152]. In summary, each spaceborne monitoring technology has its strengths and limitations.

Therefore, we should integrate multiple spaceborne techniques to explore the comprehensive process of lake ice dynamics. For example, combined satellite imagery with altimetry data can provide a comprehensive overview of the lake ice cover and thickness, compared with microwave radiometry, which gives insights into the freeze–thaw status. Further exploration of InSAR will be optimal for lake ice thickness retrieval. It is vital to improve the property retrieval of snow on lake ice both for modeling backscatter with radiative transfer models (e.g., Snow Microwave Radiative Transfer [116]) and for predicting lake ice changes with models [285].

Lake ice and its coverage duration can be identified by spaceborne sensors with frequent revisiting intervals. Compared with the threshold-based methods, the automatic

identification of lake ice (e.g., segmentation methods, unsupervised classification) is widely used. Significantly, the application of machine learning may help improve the accuracy of lake ice identification with physical models. Furthermore, the combination of remote sensing and lake ice models/physical models is promising to quantitatively understand lake ice cover and thickness, ice freeze–thaw timing, and the impact of snow cover on lake ice.

Many challenges remain when observing cryospheric hazards in the mountains. First, freely available optical images are limited to cloud-free days. Second, poor temporal availability and steep terrain limit the application of SAR detection in snow/ice avalanches. Third, the high cost of high or very-high-resolution images, except for PlanetScope images (https://www.planet.com accessed on 25 February 2024), restrict their broad usage in hazard studies. Fourth, hazards mapping, zoning, i.e., the division into districts to define potential exposure to hazards, risk evaluation, robust forecasting, cascade hazards warning, adaptation solutions, and management and mitigation are difficult due to inadequate frequency revisits of high-resolution spaceborne images, minimal corroborating field-based observations, and knowledge gaps including the complex instability and process chains of paraglacial, glacial and periglacial circumstances in rough terrain mountains further complicate cryospheric modeling.

Historically, remote sensing techniques have significantly contributed to investigating the surface features of the cryosphere and related hazards. The promising studies are to integrate advanced remote sensing, in situ observations, and modeling for a comprehensive and quantitative understanding of glaciers, frozen ground conditions, and changes, lake ice phenology, snow cover, and related hazards together with other climatic–cryospheric–hydrological components such as temperature, precipitation, runoff, and lakes. However, insufficient in situ measured data lead to difficulties in verifying the accuracy of a model's or satellite-retrieved products. Additionally, considerable challenges and substantial uncertainties remain due to the limited techniques available for accurate subsurface observations, geolocation issues when integrating datasets of different spatial resolutions (e.g., DEMs not fitting to the satellite data), and the significant manual workload in reviewing the classified results after automatic algorithms have been applied, insufficient calibration from automated retrievals for the created products, and knowledge gaps of mechanisms or triggers in a complicated environment, involving the complex topography, and rapidly changing climate in high altitudes.

## 6. Conclusions

The cryosphere changes rapidly, leading to frequent ice and rock collapses/avalanches/landslides and cascading hydrological hazards in HMA. Therefore, we reviewed the satellite-based cryosphere change studies (i.e., glacier, snow, permafrost, lake ice phenology, and glacier-related hazards), cryospheric related hydrological models, and contributions to streamflow/lakes.

The satellite-based cryospheric studies and diverse methodologies applied over the past 60 years demonstrate the practical applications for quantifying cryospheric changes, i.e., glacier coverage, geodetic glacier mass change, snow cover, SWE, soil moisture, freeze and thaw of permafrost, thermokarst lakes, lake ice phenology, GLOFs, and glacier-related hazards, as well as the potential for retrieving ground ice changes at large spatial scales. Vast quantities of observed results have been achieved in glacier mass loss and snow change, thawing permafrost, the shortened lake ice duration of most lakes, and glacial lakes in HMA. Remote sensing techniques/methodologies and related data products have been widely used to model the cryospheric hydrology change in the HMA. Consequently, the recent development of cryospheric models depends heavily on remote-sensing-derived datasets. However, the limited spaceborne subsurface observation capabilities prevent integrating complex physical processes in cryo-hydrological models. Additionally, the remote sensing datasets are rarely confirmed by ground measurements, which limits their usage in large-scale cryo-hydrological modeling studies.

Substantial uncertainties/errors and data gaps remain across the remote sensing techniques. Knowledge gaps also restrict the usage of remote sensing-based techniques and related models. These include the unknown mechanism underneath the surface, inaccurate outlines of glaciers, unclear snow variation characteristics on lake ice/glaciers, knowledge gaps in triggering process chains of cascade hazards, and unknown effects of extreme weather and climate. Moreover, it is difficult to verify the accuracy of models or satellite-retrieved data in mountainous areas with rough terrain lacking in situ measurements. Consequently, numerous errors are introduced from the inconsistent estimations noted above in water storage resources, hydrological processes, and future robust projections on climate–hydrology hazards.

For the future, additional integrated and multidisciplinary long-term observations and datasets are necessary to better understand climatic–cryospheric–hydrological change across the HMA. Specific recommendations include, first, focusing on technological innovations that are promising to advance remote sensing and retrieving methodologies, maximizing data validity for cryosphere observations at ultrahigh spatiotemporal resolution and at low cost. Second, capitalize on the fusion and assimilation of big data from multisource observations, which can provide a more comprehensive long-term view of changes in glacier/snow/permafrost/lake ice/glacial lake/cryospheric hazards, combining the strengths of different remote sensing techniques and in situ investigations to create a clearer picture of ice/water dynamics. Big data can also facilitate and implement cross-verification of different independent techniques/methods. Third, recognize and encourage that multidisciplinary approaches are vital, with multiple process-based models and machine/deep learning methodologies. Ultimately, modeling should be based on data both from remote sensing techniques and field observations, which can reinforce more accurate estimations of cryosphere–hydrosphere change and predict its hydrological cascade hazards in HMA.

**Supplementary Materials:** The following supporting information can be downloaded at https://www.mdpi.com/article/10.3390/rs16101709/s1. Table S1: An overview table sorted for the products in the cryosphere from space; Table S2: Major satellites/sensors for cryosphere monitoring; Table S3: Satellite-based products of DEMs or surface elevation difference datasets; Table S4: Methods for glacier studies from space; Table S5: Major surge-type glacier inventories in HMA; Table S6: Major methods for snow and frozen ground studies from space; Table S7: Products of snow cover and frozen ground from space; Table S8: Methods for lake ice studies from space; Table S9: Results on geodetic glacier mass balance changes in HMA; Table S10: Products of lake ice studies from space; Table S11: Download linkage for datasets, database or platforms; Table S12: Acronyms used in the paper.

**Author Contributions:** Conceptualization, Q.Y., Y.W., L.L., X.Z., X.L., N.W. and L.Z. (Liping Zhu); methodology, Q.Y., Y.W. and L.L.; Software, Y.H. and L.Z. (Limin Zhai); Validation, Q.Y., Y.W., L.L. and L.D.; Formal analysis, Q.Y., Y.W., L.L. and X.Z.; Investigation, Q.Y., Y.W. and L.L.; resources, Y.Q., L.S. and T.C.; Data curation, Q.Y., Y.W., L.L., Y.H. and L.Z. (Limin Zhai); Writing—original draft preparation, Q.Y., Y.W., L.L., L.G., L.D., L.Z. (Limin Zhai) and Y.R.; Writing—review and editing, X.Z., Q.Y., Y.W., L.L., L.G., L.D., Y.R., X.L. and N.W.; Visualization, Y.H., L.Z. (Limin Zhai), N.A. and X.J.; Supervision, L.Z. (Liping Zhu) and X.Z.; Project administration, L.Z. (Liping Zhu); Funding acquisition, L.Z. (Liping Zhu). All authors have read and agreed to the published version of the manuscript.

**Funding:** This research was funded by the Second Tibetan Plateau Scientific Expedition and Research Program (STEP, Grant number 2019QZKK0202-02) and the National Natural Science Foundation of China (41831177). Yuzhe Wang was supported by the National Natural Science Foundation of China (Grant number 42271134).

**Data Availability Statement:** Not applicable.

**Acknowledgments:** We thank the German Aerospace Center (DLR) under project XTI_GLAC6924, and all field observation stations on TP for their invaluable data support. Many thanks to the Group on Earth Observations Cold Regions Initiative (GEO CRI—http://www.geocri.org accessed on

25 February 2024). We appreciate the significant time and effort of the reviewers and their assistance to improve the quality of the manuscript.

**Conflicts of Interest:** The authors declare no conflicts of interest.

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
