# Peer review of "Remote Sensing and Modeling of the Cryosphere in High Mountain Asia: A Multidisciplinary Review"

_remotesensing, doi:10.3390/rs16101709_

Round 1
Reviewer 1 Report (Previous Reviewer 1)
Comments and Suggestions for Authors
Please find my re-review in the attachment.

Comments on the Quality of English LanguageThe English is readable, but should really be checked by a native speaker. Plural/singular forms and the use of 'the' is often wrong. I hope the authors can also improve on the somewhat 'boring' writing style.
Author Response
Dear Reviewer,
We appreciate your significant time and efforts to help us improve the quality of the manuscript. Please find the detailed responses below and the corresponding revisions in track changes in the re-submitted files.

Reviewer 2 Report (Previous Reviewer 2)
Comments and Suggestions for Authors
This paper provides a multi disciplinary review for remote sensing monitoring and modelling of the cryosphere in high mountain Asia. It is recommended to be accepted.My only suggestion is to improve the resolution of all Figures.
Author Response
Dear Reviewer,
Thank you very much for taking the time to review this manuscript. Please find the detailed responses below and the corresponding updated figures in the re-submitted files.

This manuscript is a resubmission of an earlier submission. The following is a list of the peer review reports and author responses from that submission.
Round 1
Reviewer 1 Report
Comments and Suggestions for Authors
See attachment.

Comments on the Quality of English LanguageOK, but not overwhelming, several small mistakes (grammar and wording). Should be reviewed and corrected by a native speaker.
Reviewer 2 Report
Comments and Suggestions for Authors
This manuscript focuses on remote sensing and modelling of the cryosphere in High Mountain Asia, and emphasizes spaceborne observations and hydrological models from diversified techniques/methodologies. The authors reviewed the satellite based cryosphere, including glacier, snow, permafrost, and lake ice phenology change studies, related hydrological models, and contributions to streamflow/lakes, which provided important valuable reference for the related research of Cryosphere. The theoretical basis of this manuscript is sufficient, and it is well-organized. The authors can improve the quality of this manuscript based on the following comments:
(1)The text resolution of the satellite sensor in Figure 2 needs to be improved.
(2)All abbreviations that appear for the first time require a full name. It is recommended to complete the relevant introduction.
(3)The citation of reference documents in the Table should be unified, and the introduction in Tables 2 and 3 needs to be modified.
(4)The display resolution of the legend in Figure 5 needs to be improved.
(5)It is recommended to supplement the relevant applications of Chinese remote sensing sensors, such as FY series satellites.
(6) It is suggested that the authors introduce in detail the application and achievements of modern technology, e.g. machine learning and deep learning in the Cryosphere.
Reviewer 3 Report
Comments and Suggestions for Authors
Sound detailed review of remote sensing studies of HMA cryosphere.
One cryosphere element is missing which is aufeis. It is important indicator of groundwater in cyosphere, also a hazard. I suggest you add it to the manuscript.
Reviewer 4 Report
Comments and Suggestions for Authors
Edited for remotesensing-2531057
Remote Sensing and Modelling of the Cryosphere in High Mountain Asia: A Comprehensive Review BY Qinghua Ye, Yuzhe Wang, Lin Liu, Linan Guo, Xueqin Zhang, Liyun Dai, Limin Zhai, Yafan Hu, Nauman Ali, Xinhui Ji, Youhua Ran, Yubao Qiu, Lijuan Shi, Tao Che, Ninglian Wang, Xin Li, Liping Zhu*
The cryosphere has changed significantly in High Mountain Asia (HMA), especially over the past decades, leading to diversified natural hazards, and become one of the international research focuses. The manuscript has provided the comprehensive review of cryosphere monitoring and cryo-hydrological modeling from the few main factors of cryosphere including glacier, frozen ground, snow, and lake ice, and the corresponding influences. Then the manuscript has reviewed cryosphere-hydrologic effects, and emphasized the main challenges for cryosphere monitoring with satellite-based data sets. Finally the manuscript offers the principal results and data achievements from space in the Supplementary Tables, including download linkages for available products and related data platforms.
In my opinion, the interdisciplinary review presents an essential valuable reference for the Cryosphere in High Mountain Asia from space-borne observations to hydrological models with diversified techniques/methodologies, to be published in the journal. Therefore, I recommend its publication after the minor revision.
Some minor opinions are as follows.
(1) As mentioned above, lake ice is one of the few main factors of cryosphere. Could the manuscript provide one figure or table to summarize lake ice change in 2.2.5? Perhaps there are some difficulties.
(2) The figure 2 only provides the launch time of satellites for cryosphere monitoring. Could the figure also offer the end time or last time with lines instead of circles?
(3) In my opinion, the title is not consistent with the manuscript content completely. Could the title change into Remote Sensing Monitoring and Modelling of the Cryosphere in High Mountain Asia: A Comprehensive Review?
(4) In the main manuscript, the abbreviations should have the full names when occurring firstly (such as HMA in Line 69).
(5) The words in Figure 2 are too indistinct to be recognized clearly. Please enlarge the font size, or change font.
(6) The words in Figures 3 4 and 5 are also too indistinct. Perhaps the manuscript should change the output mode.